# Thresholds for estuarine compound flooding using a combined hydrodynamic–statistical modelling approach

**Charlotte Lyddon**[1], **Nguyen Chien**[2], **Grigorios Vasilopoulos**[3], **Michael Ridgill**[4], **Sogol Moradian**[5], **Agnieszka Olbert**[5], **Thomas Coulthard**[3], **Andrew Barkwith**[6], **and Peter Robins**[4]

[1]Department of Geography and Planning, University of Liverpool, Liverpool, UK
[2]School of Engineering, University of Edinburgh, Edinburgh, UK
[3]School of Environmental Sciences, University of Hull, Hull, UK
[4]School of Ocean Sciences, Bangor University, Bangor, UK
[5]Civil Engineering, University of Galway, Galway, Ireland
[6]British Geological Survey, Keyworth, Nottingham, UK

**Correspondence:** Charlotte Lyddon (c.e.lyddon@liverpool.ac.uk)

**Abstract.** Estuarine compound flooding can happen when extreme sea level and river discharges occur concurrently, or in close succession, inundating low-lying coastal regions. Such events are hard to predict and amplify the hazard. Recent UK storms, including Storm Desmond (2015) and Ciara (2020), have highlighted the vulnerability of mountainous Atlantic-facing catchments to the impacts of compound flooding including risk to life and short- and long-term socio-economic damages. To improve prediction and early warning of compound flooding, combined sea and river thresholds need to be established. In this study, observational data and numerical modelling were used to reconstruct the historic flood record of an estuary particularly vulnerable to compound flooding (Conwy, North Wales). The record was used to develop a method for identifying combined sea level and river discharge thresholds for flooding using idealised simulations and joint-probability analyses. The results show how flooding extent responds to increasing total water level and river discharge, with notable amplification in flood extent due to the compounding drivers in some circumstances, and sensitivity ($\sim 7\%$) due to a 3 h time lag between the drivers. The influence of storm surge magnitude (as a component of total water level) on the flooding extent was only important for scenarios with minor flooding. There was variability as to when and where compound flooding occurred; it was most likely under moderate sea and river conditions (e.g. 60th–70th and 30th–50th percentiles) and only in the middle-estuary zone. For such cases, joint-probability analysis is important for establishing compound flood risk behaviour. Elsewhere in the estuary, either the sea state (lower estuary) or river flow (upper estuary) dominated the hazard, and single-value probability analysis is sufficient. These methods can be applied to estuaries worldwide to identify site-specific thresholds for flooding to support emergency response and long-term coastal management plans.

## 1 Introduction

Estuaries are the most dynamic coastal systems – crucial for global water and nutrient cycling and the biodiversity of natural habitats – and provide ecosystem services such as food security and tourism that shape the livelihoods and well-being of their communities (Barbier et al., 2011). They hold strategic value for world trade, supporting haulage and fisheries, with significant growth opportunities, e.g. in marine energy. About 60 % of the world's population lives along coastal and estuarine zones (Lindeboom, 2020), and 36 % of the UK lives within 5 km of the coast (Census, 2020). Each year people make over 270 million recreational visits to UK coasts (Elliott et al., 2018) and generate GBP 17.1 billion in tourist spending (NCTA, 2023). Sea level rise and changing storm patterns, along with the intensification of human activity in and around estuaries, e.g. littoralisation, farming, and

water management, mean estuarine communities are increasingly vulnerable to the impacts of extreme events – of which in the UK flood hazards are rated as the second-highest risk for civil emergencies, after pandemic influenza (HM Government, 2020; EA, 2023).

Estuaries are at the interface of marine (tide, storm surges, waves), hydrological, and terrestrial (precipitation causing river discharge, runoff, snowmelt, groundwater) physical processes, which interact over a range of temporal and spatial scales (Chilton et al., 2021). Standard terms follow the definitions outlined in Pugh (1996) and Chow et al. (1988). Flooding can occur when one or several of these processes cause water levels to exceed a critical threshold, such as a sea defence (EA, 2020). A threshold represents a meteorological, river, and/or coastal condition at which the flooding hazard increases (Sene, 2008). If a forecasted storm event could exceed the threshold, then action to mitigate the hazard should be taken, for example, by issuing a flood warning. In the UK, coastal flooding has an annual cost of up to GBP 2.2 billion for flood management and emergency response (Penning-Rowsell, 2014). Estuaries are particularly vulnerable to the effects of compound flood events when coastal and fluvial drivers can occur concurrently or in close succession to generate flooding (Svensson and Jones, 2004; Couasnon et al., 2020; Bevacqua et al., 2020; Robins et al., 2021). High sea levels can occur due to astronomically high spring tides and can be further exacerbated when they co-occur with storms generating large surges and waves at the coast. Alongside this, storms can generate heavy precipitation and lead to high fluvial and pluvial flows, increasing flood hazards within estuaries (Ward et al., 2018). A compound event caused devastating flood impacts in Lancaster, NW England, following Storm Desmond (4–6 December 2015), due to extended heavy rainfall and river discharges coinciding with an incoming tide (Ferranti et al., 2017).

Statistical analyses of long-term data, e.g. from paired coastal and riverine gauge observations, can show dependence between these drivers (Hendry et al., 2019; Camus et al., 2021; Lyddon et al., 2022) and can be used to examine the joint exceedance probability of estuary water levels based on when marine and terrestrial drivers are above the predefined thresholds (e.g. 95th or 99th percentile) (Kew et al., 2013; Salvadori et al., 2016). Estuaries on the western coast of Great Britain are more likely to experience co-dependent extreme events and compound flooding than those on the eastern coast, due to the prevailing southwesterly storm tracks that can bring extreme storm surges and concomitant rainfall – with the generally short and mountainous western-coast catchments causing river flows to increase quickly and coincide with the surge (Haigh et al., 2016). Beyond the floods in Lancaster, NW England, Storm Desmond caused severe compound flooding across several estuaries of western and southwestern Great Britain, amounting to over GBP 500 million TSI in flood-related damages (Bilskie and Hagen, 2018; Matthews et al., 2018). Flooding in estuaries on the eastern coast of Great Britain is more likely to be driven by independent surge and rainfall events because the catchments tend to be larger with slower runoff times and easterly storms tend not to be coupled with heavy rainfall (Svensson and Jones, 2004), although the generally longer durations of high river flows (e.g. several days for the Humber, NE England) increase the chances of high discharge coinciding with high sea levels from a separate storm. Modelling studies have shown the likelihood and impacts of compound flooding at local (Robins et al., 2021) and national scales (Ganguli and Merz, 2019; Eilander et al., 2020; Feng et al., 2023; Eilander et al., 2023) but do not specify driver thresholds that lead to compound flooding and spatial variability in the flooding of different driver combinations.

Defining critical driver thresholds for estuary flooding is crucial for the early detection and forecasting of flood events for issuing timely warnings, for operational purposes such as emergency response, and for identifying vulnerable areas to focus intervention and coastal management strategies (EA, 2009). Early warning systems and appropriate planning measures are the most widely used and reliable tools to ensure community preparedness (Alfieri et al., 2012). Early warning systems and subsequent responses require a thorough understanding of hazard behaviour and classification, and knowing when a specific environmental condition will be passed to cause flooding is vital in this framework (Šakić Trogrlić et al., 2022). Terrestrial-driven floods and marine-driven floods are generally considered separately in operational flood risk assessments (e.g. CoSMoS, Coastal Storm Modeling System, for the USA from the USGS), and there is currently a UK government policy gap in terms of estuary flood risk (EA, personal communication, 2023). Flood assessments show when a critical threshold is exceeded to cause either fluvial or coastal flooding but do not consider compound events. Modelling statistical and probabilistic methods can contribute to an understanding of the unique response of each estuary to flood drivers, where catchment typology, tidal regime, and estuary characteristics influence the behaviour of the hazard. The same water level return period at a location within an estuary can be caused by different drivers and cause different flood extents, showing the importance of understanding a range of site-specific, compound event scenarios alongside their joint probability (Olbert et al., 2023).

This research aims to identify the coastal and fluvial conditions that lead to flooding in an estuarine system. The research uses a combination of historic records of flooding, instrumental data, statistical analyses, and numerical modelling tools to identify the combined driver thresholds which cause flooding and which areas within the estuary are vulnerable to the compounding effects. The research is applied to the Conwy estuary, North Wales (N Wales), as an example of a mountainous, flashy catchment on the western coast of Great Britain which is vulnerable to the effects of storm-driven, compound flooding. The case study and methodology are described in Sect. 2, demonstrating how historic records

of flooding are supplemented with online sources, instrumental data from a paired river–tide gauge, and results from an inundation model (Sect. 3). Joint probabilities are assigned to coastal and fluvial conditions before results are considered in the context of wider flood hazard policy to improve the accuracy of flood records and flood hazard assessments in the context of future climate change and land use change for improved resilience of coastal communities (Sect. 4).

## 2 Methods

### 2.1 Conwy estuary, North Wales

The Conwy estuary is a steep and mountainous catchment in N Wales that has been shown to be one of the most vulnerable in Great Britain to compound events of extreme surges coinciding with extreme river flows (Lyddon et al., 2022). The estuary is macrotidal, which is common for the UK, with a 4–6 m tidal range. The semi-diurnal tide displays pronounced tidal asymmetry, characterised by short, fast flood tides and longer, slower ebb tides, which is typical of many macrotidal estuaries. Current speeds reach $1.3\,\mathrm{m\,s^{-1}}$ during the 2.75 h flood, while ebb current speeds are 25 %–30 % slower (Jago et al., 2024). The estuary is subject to the effects of surge-generating, low-pressure Atlantic storms, elevating sea level up to 1.6 m above predicted levels. The towns of Llanrwst in the upper estuary and Conwy and Llandudno in the lower estuary are vulnerable to this hazard, and communities, businesses, and transport networks are affected by several floods each year. Most notably, the primary road and rail network connecting North and South Wales runs through the Conwy Valley. Storm Ciara, on 9 February 2020, exemplifies the complexities of compound flooding. Ciara atypically came from the north bringing intense rainfall (80 mm in 15 h) that inundated the estuary floodplains to capacity and was held back by the rising spring tide plus 0.72 m surge. Record-breaking flows ($529\,\mathrm{m^3\,s^{-1}}$) in the main river ensued, causing widespread flooding (> 150 properties) and a "backwater effect" that flooded transport links and caused power outages. There was no warning, so residents and landowners had no chance of activating safety measures. Flooding was recorded throughout the community in local and regional news outlets (BBC, 2020; Evans, 2020; Spridgeon, 2020).

The Conwy estuary has a record of instrumental observation data available from the Cwmlanerch river gauge (https://nrfa.ceh.ac.uk/data/station/info/66011, last access: August 2023) and Llandudno tide gauge (https://ntslf.org/tgi/portinfo?port=Llandudno, last access: August 2023). River discharge recorded at Cwmlanerch is available at a 15 min temporal resolution from November 1980–February 2023, with 99 % data coverage in time. The total water level recorded at Llandudno is available at a 15 min temporal resolution from January 1994–December 2020, with 88 % data coverage in time. Total water level from the Llandudno tide gauge was linearly detrended to remove the effects of a historical sea level trend from the time series (Coles, 2001). Historic records of flooding extend back to the 1980s before the instrumental tide gauge data began; therefore tide and surge reanalysis data for this period were obtained from the Global Tide and Surge Model (GTSM). The third-generation GTSM (Kernkamp et al., 2011) has a coastal resolution of 1.25 km within Europe and is forced with meteorological fields from the ERA5 climate reanalysis to simulate extreme sea levels for the period 1979 to 2017. The tide and surge model has shown good agreement between modelled and observed sea levels and is applicable to flood risk and climate change research (Muis et al., 2016, 2020; Wang et al., 2022). The record length used in the analysis here is determined by the monitoring and modelling duration.

### 2.2 Historic records of flooding in Conwy

Natural Resources Wales (NRW) has collated information on Recorded Flood Extents to show areas that have flooded in the past from rivers, the sea or surface water, which is documented on an open-source, online data catalogue (NRW, 2020, 2023) TS2. The database of polygons (Fig. 1a) shows 22 Recorded Flood Extents in the tidally influenced the Conwy estuary. Of these Recorded Flood Extents, 14 events were driven by high sea levels or river flows or both that caused flooding by channel capacity exceedance or overtopping of defences (i.e. ignoring flooding due to obstructions, blockages, local drainage issues, and excess surface water was ignored). Instrumental river gauge data were only available for 6 of these 14 events. The behaviour of the drivers of the six recorded flood events was reconstructed from the sea level and river flow data records, including timing and magnitude of peak river discharge ($Q_{\mathrm{max}}$), total water level (TWL$_{\mathrm{max}}$), predicted tide level, and skew surge that preceded the flood (e.g. Fig. 1e and f). Figure 1c and e show the 21 November 1980 compound event where $Q_{\mathrm{max}}$ was recorded as $428\,\mathrm{m^3\,s^{-1}}$ at 03:45. TWL$_{\mathrm{max}}$ was 4.5 m at 22:00 (which included a 0.25 m skew surge); however lack of exact information on the timing of the flooding makes it difficult to determine if TWL$_{\mathrm{max}}$ contributed to flooding and whether this was a compound flood. The NRW catalogue notes that there was widespread flooding in the Conwy Valley at this time, although since this was the pre-internet era, there are no further online records. Figure 1d and f show the 26 December 2015 compound event, where $Q_{\mathrm{max}}$ was recorded as $753\,\mathrm{m^3\,s^{-1}}$ at 10:45 and TWL$_{\mathrm{max}}$ was 4.3 m at 11:00 (which included a 0.3 m storm surge). The short, 15 min time lag between $Q_{\mathrm{max}}$ and TWL$_{\mathrm{max}}$ and extreme magnitudes ($Q_{\mathrm{max}}$ was an 85th-percentile event and TWL$_{\mathrm{max}}$ was an 84th-percentile event) caused extensive flooding in Llanrwst and across the valley (ITV, 2015; Welsh Government, 2015; Jones, 2016; NRW, 2016); however, the recorded flood event in the NRW catalogue covers only a small area at Llanrwst (Fig. 1d). This suggests that historic records of flooding in the Conwy are in-

complete; hence there is a need for further information on the drivers and impacts of flooding from which to establish flood prediction patterns and thresholds. NRW identifies that the absence of a Recorded Flood Extent does not mean the area has not flooded. This information gap is expected throughout the UK.

Flood drivers $Q_{max}$ and $TWL_{max}$ during the six recorded flood events in NRW's data catalogue are shown as stars in Fig. 2. Additionally, from analysis of the $\sim 40$ years of river/sea gauge data (see Sect. 2.1), the top 50 most extreme $Q_{max}$ and corresponding $TWL_{max}$ events within a "storm window" are shown as circles in Fig. 2 (each of these corresponding events occur within a storm window of one another, defined as 20.25 h for the Conwy based on the average duration of extreme event hydrographs over a 30-year period; Lyddon et al., 2022). Gaps in the tide gauge record meant that in effect the top 72 $Q_{max}$ events were selected to identify 50 events paired with $TWL_{max}$. Similarly, the top 50 most extreme $TWL_{max}$ and corresponding $Q_{max}$ events are shown as triangles in Fig. 2. For all paired events plotted, the time lag in hours between $Q_{max}$ and $TWL_{max}$ is represented by the shape colour, and the vertical black line indicates the magnitude of the skew surge. One top 50 $Q_{max}$ event corresponded with a top 50 $TWL_{max}$ event so that 99 extreme events were identified. Not all of these 99 extreme events from the gauge records necessarily caused flooding, but these data highlight that there are potentially many events that caused flooding that are not recorded, as explored below. Further, two of the six Recorded Flood Extents corresponded with the 99 extreme events, meaning a total of 103 events are plotted in Fig. 2.

The recorded most extreme $Q_{max}$ was $901.31\,\mathrm{m^3\,s^{-1}}$, which occurred on 16 March 2019 and coincided with a $TWL_{max}$ of 6.57 m (a neap tide reaching 6.08 m combined with a 0.49 m skew surge), where there was a time lag of 3.5 h (i.e. $Q_{max}$ occurred on the ebbing tide). The relatively long time lag and less extreme $TWL_{max}$ means that this was predominantly a fluvial-driven event, rather than a compound event. Flooding was recorded across the UK including in the Conwy on this date following a particularly wet period that included two major storms, Freyer and Gareth (Met Office, 2019). The most extreme recorded $TWL_{max}$ was 8.95 m (a spring tide of 8.47 m with a skew surge of 0.48 m), which occurred on 10 February 1997 and coincided with a $Q_{max}$ of $311.52\,\mathrm{m^3\,s^{-1}}$, where there was a 1.5 h time lag (again $Q_{max}$ occurred on the ebbing tide). Whilst coastal flooding was recorded in the Conwy Tidal Flood Risk Assessment (HRW, 2008), there was no flooding recorded within the estuary, so it is not considered a compound event.

Of the top 50 $Q_{max}$ events, 39 had a time lag of $\pm 2$ h or less, of which 14 events had a time lag of $\pm 1$ h or less, showing that concurrence of $Q_{max}$ and $TWL_{max}$ has occurred regularly in the past. There was only one occasion when a top 50 $Q_{max}$ and top 50 $TWL_{max}$ co-occurred, and this event had a time lag of about an hour. Seven of the top 50 $TWL_{max}$ events

**Table 1.** Description of labels used to assign a cause of flood tag to a date.

| Label | Code |
| --- | --- |
| 0 | None |
| 1 | River discharge |
| 2 | Storm surge |
| 3 | High tide |
| 4 | Storminess |

had a time lag of $\pm 2$ h or less, of which two events had a time lag of $+1$ h or less. It is also worth noting that all top 50 $TWL_{max}$ events occurred around midday (10:30–12:15) or midnight (22:45–00:00). Spring high tides are phase-locked around midday and midnight for the Conwy region, hence increasing the chances of an extreme water level at these times.

Three standout events are circled in Fig. 2 which could be interpreted as compound events, all with extreme river discharges ($Q_{max} > 700\,\mathrm{m^3\,s^{-1}}$ and $> 77$th percentile), high total water levels ($TWL_{max} > 4$ m and $> 84$th percentile), and time lags under $\pm 1$ h. One of these three events is starred as a recorded flood event on the NRW data catalogue (26 December 2015); however, the others are not. It is important to know whether all of these extreme events in fact caused flooding as one might expect, as well as which other extreme events in the $\sim 40$-year record led to flooding, to be able to establish meaningful thresholds for flood warning.

## 2.3 Extending the record of flooding

Records of historic flood events were expanded by exploring internet records. Online resources were used to identify if flooding happened as a result of extreme coastal and/or river conditions to create a more comprehensive record of historic flood events. Web-scraping approaches (also referred to as web extraction or web harvesting) were used to evaluate whether there is further evidence of recorded flooding in the Conwy estuary within the 99 extreme $Q_{max}$ and $TWL_{max}$ events plotted in Fig. 2. The dates of all recorded extreme events were searched on DuckDuckGo, Microsoft Bing, and Google. No evidence of flooding was available for events prior to 1990; online records prior to this date are unreliable and before the "internet era". Predetermined searches specified any evidence must be for an event in the Conwy estuary from Deganwy upstream to Llanrwst (i.e. the dashed box in Fig. 1a). Train and bus cancellations were also considered evidence of flooding events. A railway line runs between Deganwy and Llanrwst, stopping at Llandudno Junction, Glan Conwy, Tal-y-Cafn, and Dolgarrog, so these stations were included in the web search. Results were supplied in browser tabs for analysis. If a date was deemed a "flooding event", the supporting evidence was investigated to see if there was any information to note the drivers of the flooding (Table 1).

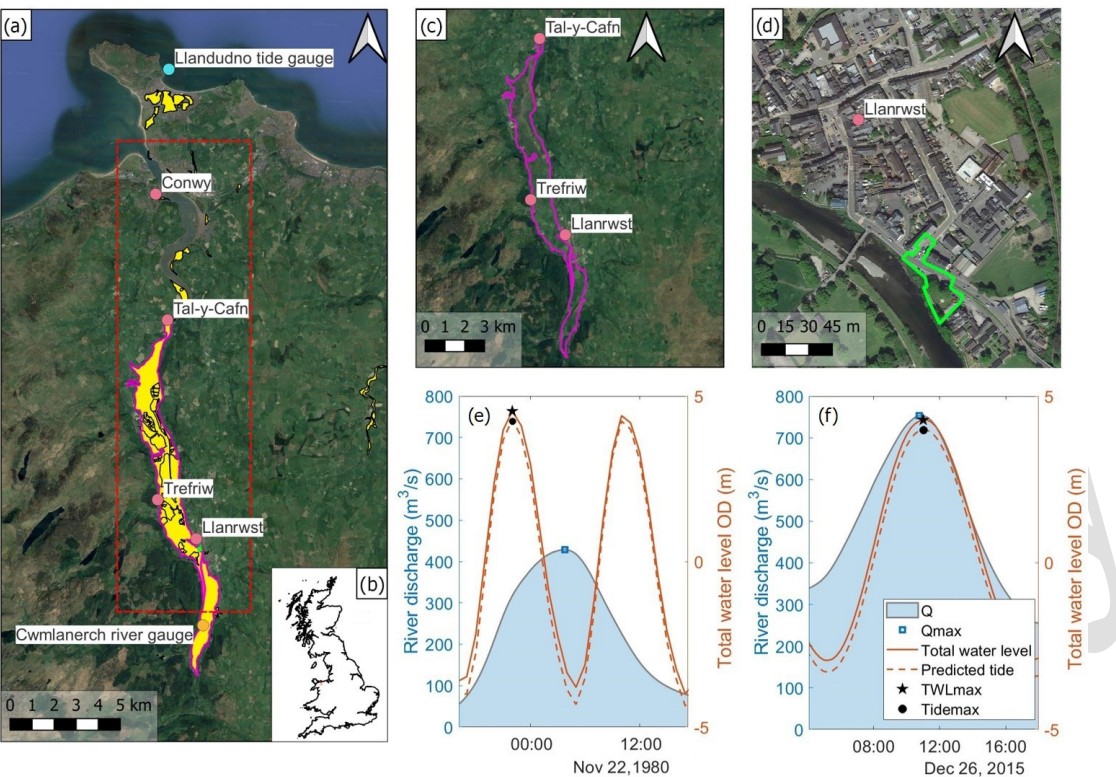

**Figure 1. (a–b)** Location and extent of all recorded flood events (yellow shading) in the region of interest (dashed red box) in the Conwy estuary, N Wales. The outlines of two recorded flood events are highlighted, 21 November 1980 (pink polygon) and 26 December 2015 (green polygon), which are shown in more detail in **(c)** and **(d)**. **(e–f)** Time series of river discharge, total water level, and predicted tide for two recorded flood events in **(c)** and **(d)**. **(a, c, d)** Basemap © OpenStreetMap contributors 2023. Distributed under the Open Data Commons Open Database License (ODbL) v1.0. OD: ordnance datum.

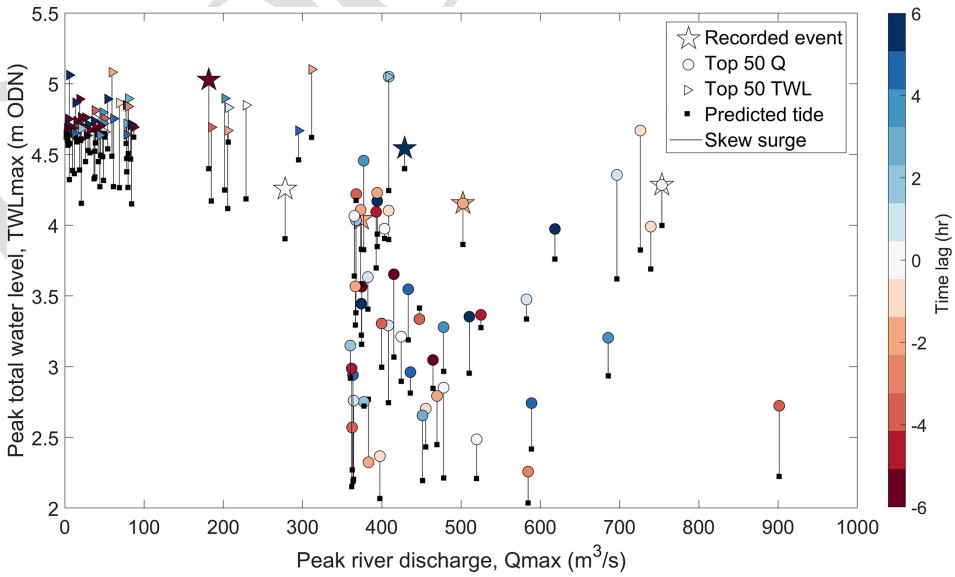

**Figure 2.** Recorded Flood Extents at Conwy (stars), top 50 $Q_{max}$ events at Cwmlanerch (circles), top 50 TWL$_{max}$ events at Llandudno (triangles), and associated predicted tide (black square) and skew surge magnitude (vertical black line) for each event. Colours indicate the length of time lag between peaks in river discharge and total water level (negative time lags indicate that $Q_{max}$ arrived before TWL$_{max}$ and so coincided with a flooding tide).

The web searches isolated an additional 26 recorded floods that matched extreme events in our analysis, as shown in Fig. 3, with yellow circles indicating these 26 events. The blue circles in Fig. 3 indicate extreme events where there was no online evidence of flooding. Labels assigned to three of the inundation events are shown in the figure. Multiple sources of evidence indicate a marine-driven flooding event on 3 January 2014, largely due to an extreme storm surge of 0.8 m, including railway cancellations, home evacuations, and road closures (Welsh Government, 2014; Sibley et al., 2015). Evidence of river-driven flooding on 16 March 2019, during Storm Gareth, was derived from news reports of damage to over 40 homes, road closures, and flood warnings issued by NRW (BBC, 2019; FloodList, 2019; Met Office, 2019). Evidence of river-driven and marine-driven flooding suggests that 9 February 2020 had a compound flood event. Figure 3 provides a more comprehensive record of flood inundation than shown in Fig. 2; however, data gaps in instrumental time series, online evidence, and what information was recorded leave uncertainty in where to set driver thresholds and patterns for flooding, especially for less extreme $Q_{max}$ and $TWL_{max}$ values that led to compound flooding.

## 2.4 Hydrodynamic inundation model

The CAESAR–Lisflood hydrodynamic model (Coulthard et al., 2013; Skinner et al., 2015; Harrison et al., 2021) was used within a sensitivity test framework to simulate a series of idealised event scenarios which represent plausible combined river and sea level conditions to identify which combination of drivers leads to flooding in the Conwy. CAESAR–Lisflood is a geomorphological and landscape evolution model that combines the LISFLOOD-FP 2D hydrodynamic flow model (Bates et al., 2010) with the CAESAR geomorphic model. Lisflood uses a flow-routing algorithm that determines the direction of flow based on the elevation gradient and conserves mass and partial momentum. CAESAR–Lisflood does not run in 3D, and this functionality is not required to explore flood inundation. Baroclinicity is not an important process to represent for this research and would require additional computational expense.

### 2.4.1 Model domain

The model domain includes the tidally influenced the Conwy estuary, downstream of the Cwmlanerch river gauge on the river Conwy and extending offshore into Conwy Bay and the Menai Strait at the coastal boundary. A number of sources were combined to generate the land elevation data required to build the model, including (a) seabed bathymetry, (b) land elevations, and (c) the location and heights of existing flood defences. The domain topography was based on the marine DEM (digital elevation model), lidar DTM (digital terrain model), and OS Terrain 5 DTM, all available through Digimap (https://digimap.edina.ac.uk/, last access: November 2022). The lidar DTM data were used to check and, where necessary, augment the flood defences vector database, obtained from the NRW data catalogue (https://datamap.gov.wales/, last access: January 2023). The processing steps undertaken to produce the model domain are described in Supplement Sect. S1 (Vasilopoulos et al., 2023).

### 2.4.2 DEM calibration

CAESAR–Lisflood was run in reach mode, in which the model is forced with discharge and water level time series at the upstream (river) and downstream (offshore) boundaries, respectively. For the upstream boundary, a time series of water discharge ($m^3 s^{-1}$) measured at the Cwmlanerch gauge was used. The dataset provided by NRW has a 15 min temporal resolution and covers the calibration period of 1 March–16 April 2021. For the offshore boundary, a time series of measured sea levels at Llandudno was used, provided by the British Oceanographic Data Centre (BODC). It contains measured levels above the Llandudno chart datum (CD) at 15 min intervals and spans the same period as the time series of discharge. The tidal water levels were converted to ordnance datum (OD) by adjusting for the vertical offset between CD and OD (i.e. −3.85 m). Manning's roughness coefficient for the river channels and marine areas was set to 0.022, the Courant number was set to 0.6, and the Froude limit was set to 0.8. To avoid water accumulation behind flood defences when overtopping occurred, a water loss function of $0.2 \, m \, d^{-1}$ was applied. The function was only applied to the floodplains to avoid affecting river or seawater levels. Only the hydrodynamic component of the model was used for the simulations described here, and simulated water levels were exported at 15 min intervals for further analysis.

Simulated water levels were compared against corresponding values obtained from gauges within the estuary at Pont Fawr, Trefriw, and Tal-y-Cafn (see Fig. 4). The gauges at Pont Fawr and Trefriw are maintained by NRW and monitor water levels at 15 min intervals, relative to OD. At Tal-y-Cafn a pressure logger was installed in October 2020 (53.23° N, 3.82° W) that also provided measured water levels, relative to OD at 15 min intervals. Initially the DEM had incorrect channel bed elevations due to the lidar shortcomings for inundated areas (further detail in Supplement Sect. S1). We approximated the correct channel bathymetry by manually adjusting the channel bed elevations, re-running the simulation, and comparing simulated and observed water levels. We repeated this process until we reached a satisfactory agreement between observed water levels and model predictions at the three gauges. With this method the bed profile is adjusted until it simulates the observed water profile taking into account flow non-uniformity (Neal et al., 2021). The calibrated DEM is shown in Fig. 4a together with the locations of the various gauges used in the study. After the final DEM adjustment (Fig. 4b), RMSE values were 0.59, 0.39, and 0.69 m (Fig. 4c–e) for Pont Fawr, Trefriw, and Tal-

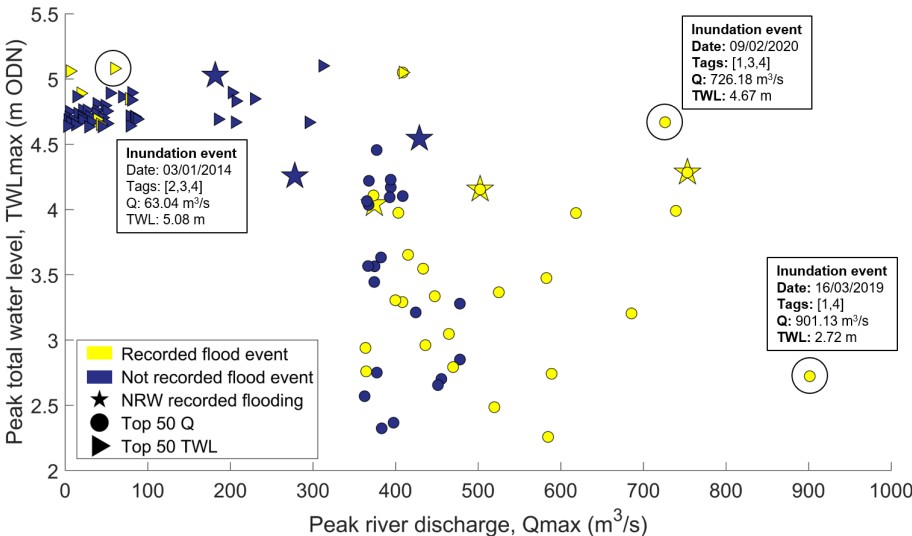

**Figure 3.** Recorded Flood Extents and top 50 $Q_{max}$ and top 50 $TWL_{max}$ events colour-coded to show those events which were inundation events (yellow) and those which were non-inundation events (blue). Three events are highlighted to show drivers, timing, and labels for the cause of flooding.

y-Cafn, respectively. Flood peaks were isolated in the calibration period, and RMSE values were 0.57, 0.19, and 0.29 m for Pont Fawr, Trefriw, and Tal-y-Cafn. Improved RMSE scores for flood peaks indicate the model is able to capture the magnitude of the largest and most prominent peaks. Higher RMSE values in the upper estuary (Pont Fawr gauge) could be attributed to the omission of tributaries in the model that flow into the Conwy downstream of the Cwmlanerch gauge (upstream boundary of the model). These inputs are, as a result, not represented in the discharge data forcing the model. Nevertheless the setup remains suitable for the purposes of this research.

## 2.5 Idealised boundary conditions for model scenarios

The idealised model scenarios were used to add more detail to the historic records of flooding and instrumental data (Figs. 2 and 3) to enable driver thresholds for flooding to be established. Three scenarios, each consisting of 520 simulations, tested the influence of the relative drivers of estuary flooding (tidal water level, storm surge, river discharge, and time lag; see Table 2 and Fig. 5). The simulations consisted of 40 river discharge conditions with incrementally increasing $Q_{max}$, in combination with (Scenario 1) 13 incrementally increasing tide levels combined with a maximum storm surge, (Scenario 2) 13 incrementally increasing tide levels combined with a mean storm surge, and (Scenario 3) 13 incrementally increasing tide levels combined with a maximum storm surge and a 3 h time lag. In total, 40 ($Q_{max}$) × 13 ($TWL_{max}$) × 3 (scenarios) = 1560 discrete simulations were performed. Each simulation was run for a period of 72 h, allowing for model spin-up (thus allowing the assumed initial condition to become consistent with the hydrodynamic

system) and with $TWL_{max}$ and $Q_{max}$ occurring after ∼ 40 h. These boundary conditions are described in more detail below.

### 2.5.1 River discharge

The following method was undertaken to generate 40 idealised discharge time series parameterised on the hydrology of the Conwy. Firstly, a two-parameter gamma distribution was used to generate a synthetic series of normalised, idealised gamma curves that represent hydrograph shapes that cover the natural range of river flow behaviours experienced in the Conwy based on 30 years of river discharge data from the Cwmlanerch river gauge (see Robins et al., 2018). The gamma curve with the gradient of the rising hydrograph limb that most closely resembled the average gradient of the top 50 $Q_{max}$ events analysed in this study was selected. The selected idealised hydrograph had the largest gradient representing the flashiest flow behaviour. The magnitude of the idealised hydrograph was then scaled to a peak discharge $Q_{max}$ of 25 m³ s⁻¹ (i.e. a relatively small river flow event that will not likely cause flooding), with a base flow of 20 m³ s⁻¹ which represents mean flow conditions. The scaling of $Q_{max}$ was successively increased from 25 m³ s⁻¹, in 25 m³ s⁻¹ increments, up to a $Q_{max}$ of 1000 m³ s⁻¹ (i.e. slightly greater than the maximum recorded event of 901 m³ s⁻¹), always keeping a base flow of 20 m³ s⁻¹. This created a realistic range of 40 river discharge event time series that were applied to all three scenarios. For each simulation, $Q_{max}$ occurred at 40 h (Fig. 5).

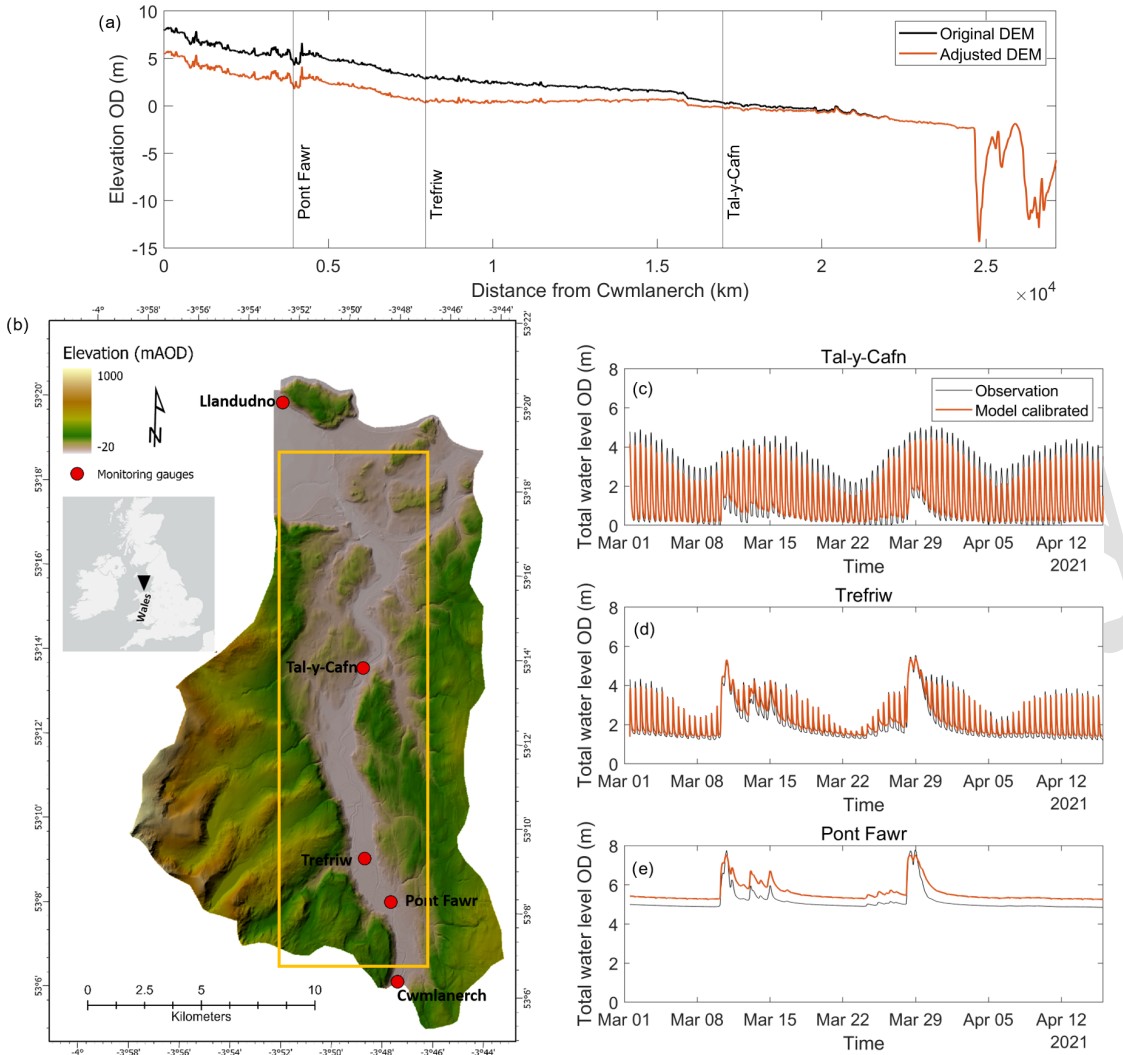

**Figure 4. (a)** Calibrated Conwy estuary model domain showing elevations relative to the ordnance datum and location of monitoring gauges. The region of interest in the estuary is shown (orange box, size of $3920 \times 19\,580$ m). **(b)** Longitudinal profile along the channel centreline showing the original elevation derived from the lidar DTM (black) and adjusted elevation (red). Comparison between observed (black) and simulated (red) time series of water levels are shown at **(c)** Pont Fawr, **(d)** Trefriw, and **(e)** Tal-y-Cafn. mAOD: metres above the ordnance datum.

### 2.5.2 Total water level

The boundary conditions for the total water level consisted of 13 time series for each of the three scenarios. These time series were created using idealised tidal signals combined with residual surges. Firstly, a sinusoidal elevation with a period of 12.42 h (equivalent to the dominant M2 tidal constituent) was created. This was parameterised to represent mean neap tides at Llandudno. Mean spring and neap tidal amplitudes and high-tide levels were determined using a harmonic analysis (T-Tide, Pawlowicz et al., 2002), based on 12 months of tide gauge data from Llandudno (2002–2003). A subsequent tidal prediction revealed that mean high-water neap tides reach 1.82 m (OD) and mean high-water spring tides reach 3.6 m (OD) at Llandudno. The elevation time series was then reproduced 13 times, each time successively increasing the amplitude so that high water was incrementally increased by 25 cm until it was equivalent to spring high tides. This experimental design purposely neglected the influence of other constituents so that the results were standardised. The model simulated the shallow-water propagation of the tide advancing up the estuary.

Secondly, for each of the three scenarios, a residual surge was added to the 13 elevation time series to represent the meteorological contribution to the total water level. The shape of the surge was representative of typical storm conditions for Llandudno (Environment Agency, 2016), as shown in Fig. 5. The surge was shifted in time so that the maximum surge

height coincided with the fourth high tide (at around 40 h). For Scenario 1 and Scenario 3, the surge was scaled to the magnitude of the maximum observed skew surge (1.03 m). The resultant 72 h time series represented several tidal cycles where flooding was not expected (tide-only), followed by a tide + max surge event at $\sim 40$ h (where the peak water level is denoted as $TWL_{max}$), before the regular tidal cycles resumed (Fig. 5a and c). For Scenario 2, the procedure was repeated, this time by applying a mean observed skew surge (0.13 m) to the predicted tide series (Fig. 5b).

### 2.5.3   Time lag

The timing of $Q_{max}$ relative to $TWL_{max}$ is a key factor in determining compound flooding hazards. This time lag was therefore considered in our sensitivity framework. From the 30-year Cwmlanerch discharge record, we calculated the distribution of time lags (following the method of Lyddon et al., 2022), as shown in Fig. 5d. Peaks in river discharge most commonly occurred 0–4 h before peaks in total water level, i.e. on the rising tide. Initially (Scenario 1 and Scenario 2), we implemented the most common time lag of 0 h (i.e. both $Q_{max}$ and $TWL_{max}$ were at 40 h as shown in Fig. 5a for Scenario 1 and Fig. 5b for Scenario 2). Next, a $-3$ h time lag was implemented as shown in Fig. 5c, since this was the next most common time lag (Fig. 5d), and applied to the 13 tide + max surge time series and 40 discharge time series (collectively named Scenario 3). In total, 13 ($TWL_{max}$) × 40 ($Q_{max}$) × 3 (scenarios) = 1560 simulations of 72 h duration were computed, as summarised in Table 2 and Fig. 5.

### 2.6   Simulations of flooding

The following methodology was applied to identify the extent of flood extent under each scenario generated in Sect. 2.5. The flooding problem can be represented as a function:

$$FloodArea = f(Q_{max}, TWL_{max}, SurgeHeight, TimeLag), \quad (1)$$

where FloodArea quantifies the inundation area (km$^2$) of the Conwy estuary floodplains, as a function of $Q_{max}$ (25–1000 m$^3$ s$^{-1}$), $TWL_{max}$ (tide + max surge) (2.25–6 m), SurgeHeight (max = 1.03 m, mean = 0.13 m), and TimeLag (0–3 h), as specified in Eq. (1).

A high-performance computing system, Supercomputing Wales (https://www.supercomputing.wales/, last access: August 2023), was used to efficiently run the CAESAR–Lisflood solver. The system is capable of handling multiple concurrent computing tasks to allow the parameter space to be partitioned into "job blocks". Blocks were submitted to the system using the Slurm (https://slurm.schedmd.com/, last access: August 2023) workload manager for batch processing. A typical 72 h simulation took 1.2–2 h of CPU runtime (on four Intel Xeon cores operating at 2.1 GHz). Overtopping of levees and shallow flows over floodplains can lengthen the

computational time, while dry parts of the catchment do not affect the computing time.

The output data comprise water depth grids in time layers with an interval of 15 min. Only data of time layers between 2300 and 3500 min ($\sim$ 38–58 h), corresponding to the period of widest flooding extents, were stored to reduce space. Post-processing to summarise outputs and calculate FloodArea was completed remotely to reduce the transfer load from the nodes to the local computer.

### 2.7   Scenario analysis

An initial baseline "no-flooding" simulation was performed, from which to calculate FloodArea in all subsequent simulations. The baseline simulation represented moderate river flow and sea level conditions, whereby water was contained within the main channel, with dry floodplains, and high water levels submerged mid-channel shoals. The baseline was drawn from an actual event on 27 January 2016, in which no inundation occurred. This case approximates the Scenario 1 simulation ($Q_1 TWL_3$) (i.e. $Q_{max} = 25$ m$^3$ s$^{-1}$, $TWL_{max} = 3.7$ m). A mask has been used to define the region of interest (ROI) (see Fig. 1a), an area of 196 × 979 cells or $\sim 7.7$ km$^2$, which encompasses the estuary floodplains from the tidal limit at Cwmlanerch to the Conwy Tunnel near the estuary mouth. Six mid-channel shoals were excluded with areas ranging from 0.003 to 0.17 km$^2$. The baseline scenario comprises 13 982 wet cells in this ROI ($\sim 5.59$ km$^2$). For each simulation, the maximum total flooded area in the ROI was recorded, from which the baseline no-flooding wet area was subtracted to create the simulated FloodArea. A floodplain model cell was considered to have flooded when the local water level exceeded a threshold of 2.5 cm. Wetted surfaces need some time to drain, hence the variation in flooded areas lags behind the water level variations. Furthermore, the minima of the flooded areas do not fully develop before the next flooding phase occurs. As experimented with a number of scenarios accompanying the study, if the depth threshold was set to zero, any thin layer of water would be considered inundation and then the flooded area would monotonically increase (not shown here). Once the land is wet there is no way to change back into dry. Only new events with higher water levels may expand the inundated area. This is a practical decision, but we also realise that the flooding area is relatively insensitive when this depth threshold varies from 2.5 to 12.5 cm. The FloodArea for each simulation was the inundated area exceeding this threshold. The FloodArea and absolute difference in FloodArea (between scenarios) are presented throughout the 520-simulation parameter space for each of Scenarios 1–3.

Spatial inundation maps were presented. Four cases were presented in this way, based on the Scenario 3 simulations: (i) TWL-dominated flooding, (ii) $Q$-dominated flooding, (iii) moderate compound flooding, and (iv) extreme combined flooding. Spatial variability in flooding was also pre-

**Table 2.** Summary of model scenarios, each containing 520 combination simulations.

| Set of 520 combination simulations | Peak total water level (TWL$_{max}$) | River ($Q_{max}$) | Time lag |
|---|---|---|---|
| Scenario 1 | (Neap : 25 cm : spring) + max surge = 1.03 m | 25 : 25 : 1000 m$^3$ s$^{-1}$ | 0 h |
| Scenario 2 | (Neap : 25 cm : spring) + mean surge = 0.13 m | 25 : 25 : 1000 m$^3$ s$^{-1}$ | 0 h |
| Scenario 3 | (Neap : 25 cm : spring) + max surge = 1.03 m | 25 : 25 : 1000 m$^3$ s$^{-1}$ | −3 h |

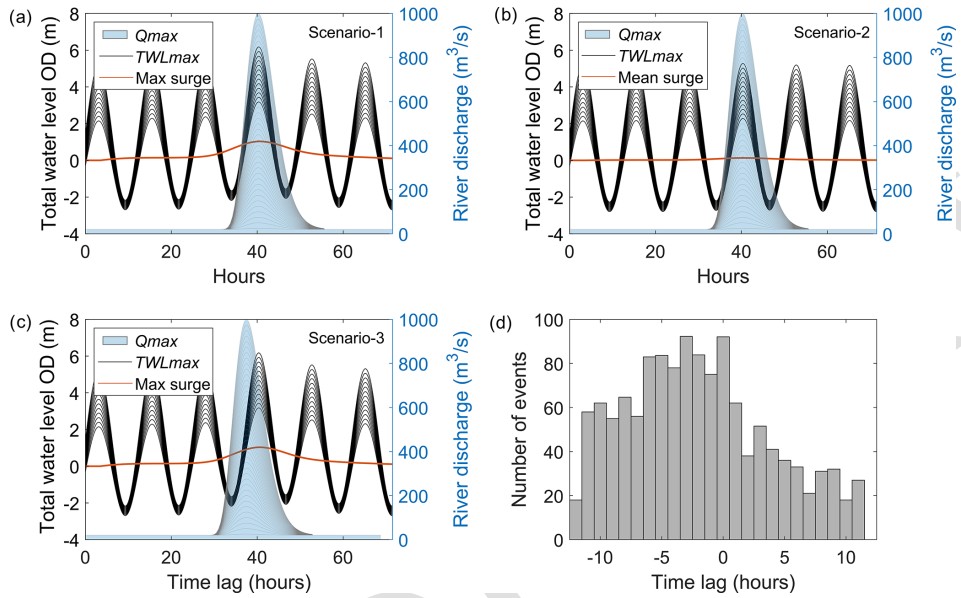

**Figure 5.** Idealised model boundary conditions for **(a)** Scenario 1, **(b)** Scenario 2, and **(c)** Scenario 3. Sea levels comprised **(a)** tide + max surge with a 0 h time lag (at ∼ 40 h), **(b)** tide + mean surge with a 0 h time lag, and **(c)** tide + max surge with a −3 h time lag. Each scenario in **(a)**–**(c)** also shows 40 river discharge hydrographs with a baseflow of 20 m$^3$ s$^{-1}$ and each with a successively increased river flow event with $Q_{max}$ occurring at ∼ 40 h. **(d)** Histogram of recorded time lag values between all $Q_{max}$ values at Cwmlanerch and TWL$_{max}$ values at Llandudno, spanning the period 1980–2023.

sented as variations in lateral flood extent (m) across east–west transects of the floodplains at regular 20 m intervals, from the estuary mouth to the tidal limit – done this way since the Conwy is almost aligned in the north–south direction (typical deviation in angle of ±30°). Again, the four cases (i–iv) above were presented in this way for lateral flood extent, based on the Scenario 3 simulations. For each case (i–iv), three simulations were presented with a similar Flood-Area: (i) TWL-dominated, 3.1–6.5 km$^2$; (ii) $Q$-dominated, 11.13–11.8 km$^2$; (iii) moderate compound, 5.4–8.3 km$^2$; and (iv) extreme compound, 8.8–9.1 km$^2$.

## 2.8 Estimating joint probabilities

Joint probabilities are important in statistics, providing a way to model and analyse the simultaneous occurrence of events. In the context of flood analysing, the joint probabilities identify the likelihood of combinations of coastal and river conditions occurring and capture relationships between variables (Wu et al., 2021; Olbert et al., 2023; Moradian et al., 2023).

The joint probability of river and sea level conditions can be interpreted in the context of (i) hydrodynamic-model outputs to identify the likelihood of combinations of conditions occurring to create a flood hazard and (ii) recorded historic flood events to provide context to the severity of flood events. Copulas are effective at modelling non-linear dependence structures and joint distribution between two variables. The copula functions (Sklar, 1959) are used here to generate synthetic bivariate pairs of extreme sea levels and river discharges, thus making their respective probability distribution more robust to applying joint-probability methods. The copula method was employed in this study to compute joint probabilities for extreme sea levels and river flows co-occurring in the Conwy for the first time. The joint probabilities were computed using the framework introduced by Sadegh et al. (2017) and Moradian et al. (2023). The proposed framework uses three main components: (i) 16 statistical distributions were employed to identify the best marginal distributions; (ii) 26 distinct copula functions were applied to sea level and river flows; and (iii) the Bayesian method

was employed to compute the joint probabilities. The following sections provide a concise overview of the steps involved in this framework, while more comprehensive details can be found in Sadegh et al. (2017, 2018), Yazdandoost et al. (2020), and Moradian et al. (2023).

### 2.8.1 Statistical marginal distributions

To identify the most suitable marginal distributions for the data, researchers commonly employ parametric or nonparametric distributions. It is important to note that each variable's marginal distribution is modelled using the best-fitted distribution, as shown in Table 6 of Moradian et al. (2023). To assess the accuracy of the marginal distributions, their significance at a 5 % level is evaluated using the chi-squared goodness-of-fit test (Greenwood and Nikulin, 1996). Furthermore, various metrics are used for statistical evaluations, as detailed in Table 5 of Moradian et al. (2023). These metrics include the Akaike information criterion (AIC), Bayesian information criterion (BIC), maximum likelihood estimation (MLE), Nash–Sutcliffe efficiency (NSE), and root mean square error (RMSE).

### 2.8.2 The copula method

Copula functions are mathematical functions that link or connect time-independent variables (Nelsen, 2007), irrespective of their individual distribution characteristics (Genest and Favre, 2007). According to Sklar's theorem (Sklar, 1959), if we have two continuous random variables $x$ and $y$ with probability density functions of $f_x(x)$ and $f_y(y)$ and cumulative distribution functions of $F_x(x)$ and $F(x)$, respectively, and if both and have the same marginal distribution function $F$, then there exists a unique copula function, $C$ of $[0.1]^2 \rightarrow [0.1]$, which serves as a bivariate cumulative distribution function and has uniform margins of

$$F(x, y) = C(F_x(x), (F_x(y)). \tag{2}$$

In an $n$-dimensional space, the cumulative distribution function $F$ can be defined in terms of the copula function $C$ and the marginal distribution functions as follows:

$$F(x_1 x_2, \ldots, x_n) = C(F_1(x_1), F_2(x_2), \ldots, F_n(x_n)), \tag{3}$$

where $F_1, F_2, \ldots, F_n$ are the marginal distribution functions (Nelsen, 2007).

A wide range of copula functions are available, categorised into various families such as Gaussian, Plackett, Archimedean, elliptical, and $t$ families (Abbasian et al., 2015). Table 4 in Moradian et al. (2023) provides a compilation of the applied 26 copula families and their corresponding mathematical descriptions. Here, to choose the best copula family, different metrics were used according to Table 5 in Moradian et al. (2023). In addition, the correlation coefficients for the used flood pairs are Pearson's linear correlation

coefficient, Kendall's tau correlation coefficient, and Spearman's rho correlation coefficient (Akoglu, 2018).

The statistical method entails assessing the likelihood of an event, taking into account existing knowledge of conditions that may be associated with the occurrence of the event. The concept has demonstrated remarkable success in diverse fields, including hydrology (Sadegh et al., 2017) and weather forecasting (Khajehei et al., 2017; Yazdandoost et al., 2020).

## 3 Results

Results are presented for the simulated FloodArea for Scenarios 1–3 in the Conwy estuary (Sect. 3.1–3.3), where a range of 1560 idealised simulations represent likely sea level and river flow "compound storm events" that could lead to flooding. Next (Sect. 3.4), for Scenario 3, a selection of simulated flooding maps and along-channel flooded width graphs are presented. Finally (Sect. 3.5), joint probabilities are assigned to the compound flood drivers.

### 3.1 Scenario 1 (tide + max surge combined with river discharge series and a 0 h time lag)

For Scenario 1, a surge tide event (skew surge = 1.03 m) was simulated, with a 0 h time lag (i.e. $Q_{max}$ and $TWL_{max}$ occurred simultaneously at 40 h of the 72 h simulations). The simulated FloodArea (km$^2$) for all 520 simulations is shown in Fig. 6, where white represents little to no flooding and red indicates the maximum flood extent ($> 10$ km$^2$). The top 50 $Q_{max}$ and $TWL_{max}$ events, as well as the recorded flooding events, are also shown. As expected, there was no or little ($< 1$ km$^2$) flooding simulated under the low-magnitude river flow and sea level events ($Q_{max} < 100$ m$^3$ s$^{-1}$ and $TWL_{max} < 4$ m). Flooding was not simulated with a $Q_{max}$ of 25 m$^3$ s$^{-1}$ until $TWL_{max}$ was 3.95 m, and then as $Q_{max}$ was increased, a reduced $TWL_{max}$ was needed to cause flooding. For example, flooding was simulated with $Q_{max} = 50$ m$^3$ s$^{-1}$ and $TWL_{max} = 3.6$ m, as well as $Q_{max} = 100$ m$^3$ s$^{-1}$ and $TWL_{max} = 3.4$ m. FloodArea increased as $Q_{max}$ and $TWL_{max}$ increased. The simulated maximum FloodArea was 11.2 km$^2$ under the $Q_{max} = 1000$ m$^3$ s$^{-1}$ and $TWL_{max} = 10$ m combination.

The contours shown in Fig. 6 connect the model simulations with a similar FloodArea (although not necessarily inundation of the same areas within the floodplains) and suggest a complex relationship between $Q_{max}$ and $TWL_{max}$ drivers in terms of simulated flooding. The contour gradients, shapes, and separation can therefore be interpreted to explain the dynamics of flooding. The contour gradients change across the range of simulations as FloodArea becomes more or less sensitive to one driver or the other. The 1 and 2 km$^2$ contours are broadly straight diagonals (bottom left part of Fig. 6), as are the 9, 10, and 11 km$^2$ contours (top right part of Fig. 6). In these cases, FloodArea is broadly equally

sensitive to both $Q_{max}$ and $TWL_{max}$ drivers. Convex contours (e.g. the middle sections of the 3 and $4\,km^2$ contours in Fig. 6) indicate a compounding flood effect, as the addition of both drivers amplifies FloodArea. Conversely, concave contours (e.g. the middle sections of the $5$–$7\,km^2$ contours in Fig. 6) indicate a degressive flooding effect, where the combination of the drivers leads to relatively less FloodArea. There is a widening between the convex ($4\,km^2$) and concave ($5\,km^2$) contours in the centre of Fig. 6, indicating that simulated flooding was relatively insensitive to changes in $Q_{max}$ between 350 and $500\,m^3\,s^{-1}$ and $TWL_{max}$ between 4 and 5 m. Hence, several simulated compound event permutations within these driver ranges produced broadly a similar FloodArea. Contours that are near horizontal (e.g. the 5 and $6\,km^2$ contours in the top left and middle parts of Fig. 6) indicate that changes in flooding are predominantly driven by changes in $TWL_{max}$. Contours that are nearly vertical (e.g. the 5 and $6\,km^2$ contours in the bottom middle part of Fig. 6) indicate that changes in flooding are predominantly driven by $Q_{max}$. Contours that are relatively close together (e.g. $5$–$7\,km^2$ contours, where $TWL_{max} > 5.25\,m$) potentially indicate key thresholds where small changes in one or both drivers lead to large changes in flooding.

### 3.2 Scenario 2 (tide + mean surge combined with river discharge series and a 0 h time lag)

Scenario 2 simulated the effect on flooding of a mean surge magnitude, in opposition to the maximum surge simulated in Scenario 1. The difference from Scenario 1 in simulated FloodArea is shown in Fig. 7, by subtracting FloodArea results of Scenario 2 from Scenario 1. The $TWL_{max}$ boundary conditions were lower for Scenario 2 (2.25–5.25 m) than for Scenario 1 (3.75–6.25 m), due to the smaller contribution of the surge, giving insight into flooding dynamics under lower $TWL_{max}$ values. Both sets of scenarios have the same underlying M2 tidal signal, so the absolute difference in FloodArea is due to the influence of the surge magnitude/shape for each scenario. All Scenario 1 simulations cause a larger FloodArea than Scenario 2 simulations for the same $Q_{max}$ and $TWL_{max}$ values. The influence of the different surge magnitudes/shapes on FloodArea has the greatest impact under high $TWL_{max}$ conditions ($> 4.25\,m$) and with $Q_{max}$ values below $500\,m^3\,s^{-1}$, causing a variance of up to $5\,km^2$ in FloodArea. Under scenarios of a low river and low sea level (bottom left of grid) or high river and sea level (top right of grid), a larger surge consistently causes 2–3 km² of more FloodArea.

### 3.3 Scenario 3 (tide + max surge combined with river discharge series and a −3 h time lag)

Scenario 3 simulated the effect on the flooding of a −3 h time lag between $Q_{max}$ and $TWL_{max}$, in opposition to the 0 h time lag simulated in Scenario 1 (both scenarios simulated a maximum surge event). Differences in FloodArea under an assigned −3 h time lag (i.e. $Q_{max}$ preceding $TWL_{max}$ by 3 h, hence occurring during flooding tide), compared with Scenario 1, are shown in Fig. 8. Generally, a similar trend in flooding was simulated for both scenarios and the gradients of the FloodArea contours were similar (see also Fig. S2 in the Supplement). One interesting difference, however, was that lower-magnitude drivers ($Q_{max} < 200\,m^3\,s^{-1}$, $TWL_{max} < 3\,m$) simulated a larger FloodArea for Scenario 3 than Scenario 1. The FloodArea contours in Scenario 3 were smoother in shape than for Scenario 1, most notably on the 5 and $6\,km^2$ contours. This could indicate a more compounding effect of the drivers with a −3 h time lag, since the lag causes more of the river water on the rising limb of the hydrograph to be retained within the estuary by the flooding tide. The simulated FloodArea was sensitive to the shift in time lag, however, with notable variation depending on simulations. The blue cells in Fig. 8 indicate that the scenarios with a −3 h time lag produced a greater FloodArea than in Scenario 1. The −3 h time lag had a small influence (generally $< 0.5\,km^2$) on FloodArea for $Q_{max} < 425\,m^3\,s^{-1}$ across all $TWL_{max}$ simulations. For $Q_{max} > 425\,m^3\,s^{-1}$, the differences in FloodArea were generally $> 0.5\,km^2$. The greatest difference in FloodArea was $1.2\,km^2$ from the simulation with $Q_{max} = 475\,m^3\,s^{-1}$ and $TWL_{max} = 4.7\,m$. Differences in FloodArea $> 1\,km^2$ were also simulated for $Q_{max} = 550$–$650\,m^3\,s^{-1}$ and $TWL_{max} < 5\,m$. For $TWL_{max} > 5\,m$ and $Q_{max} > 800\,m^3\,s^{-1}$, FloodArea appeared less sensitive to the time lag (differences of $< 0.5\,km^2$). However, for $TWL_{max} < 5\,m$ and $Q_{max} > 800\,m^3\,s^{-1}$, FloodArea appeared more sensitive to the time lag (differences of 0.5–1 km²), presumably because the stronger river discharges were able to counter the blocking effect of weaker tidal currents. Irrespective of the time lag, a $Q_{max}$ of 475–600 m³ s⁻¹ was again shown as the river conditions where there is a marked change in FloodArea and high sensitivity to $Q_{max}$. A −3 h time lag produces a 7.7 % increase in flooding across the parameter space compared with Scenario 1; Scenario 1 produced a total of $3299\,km^2$ FloodArea, and Scenario 3 produced $3553\,km^2$ FloodArea.

### 3.4 Spatial distribution of the flooded area

Aside from simulating the FloodArea considered in Sect. 3.1–3.3, it is also important to specify where the simulated flood water is distributed. To quantify the distribution of flooding in various parts of the estuary-catchment system, four cases were considered.

a. *TWL-dominated*. $TWL_{max} \geq 6.1\,m$, $Q_{max} \leq 25\,m^3\,s^{-1}$.

b. *Q-dominated*. $TWL_{max} \leq 3.1\,m$, $Q_{max} \geq 1000\,m^3\,s^{-1}$.

c. *Moderate compound*. $TWL_{max}$ 4.7–4.9 m, $Q_{max}$ 475–500 m³ s⁻¹.

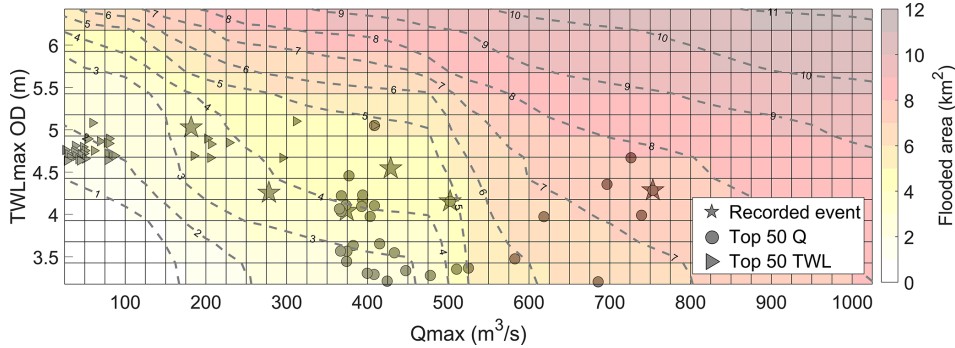

**Figure 6.** Scenario 1 (13 tide + max surge water levels combined with 40 river flow events, with a 0 h time lag): coloured surfaces represent modelled FloodArea (km$^2$) from combinations of 520 $Q_{max}$ and TWL$_{max}$ simulations. The contours link common FloodArea magnitude. Shapes correspond with Fig. 2 and indicate extreme $Q_{max}$ and TWL$_{max}$ values within the historical record (NRW recorded flood events, stars; top 50 TWL$_{max}$, triangles; top 50 $Q_{max}$, circles).

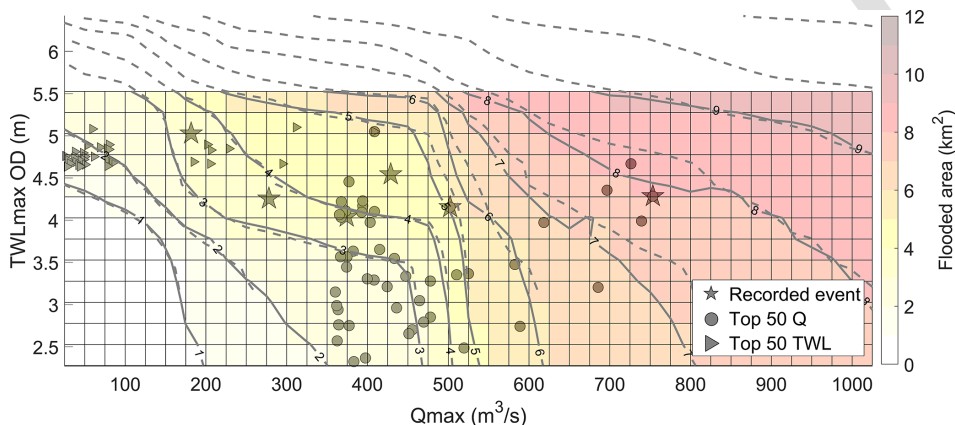

**Figure 7.** Scenario 2 (13 tide + mean surge water levels combined with 40 river flow events, with a 0 h time lag): coloured surfaces represent modelled FloodArea (km$^2$) from combinations of 520 $Q_{max}$ and TWL$_{max}$ simulations. The dashed contours link common FloodArea magnitude for Scenario 2, whereas the solid contours refer to Scenario 1 for comparison. Shapes correspond with Fig. 2 and indicate extreme $Q_{max}$ and TWL$_{max}$ values within the historical record (NRW recorded flood events, stars; top 50 TWL$_{max}$, triangles; top 50 $Q_{max}$, circles).

d. *Extreme combined.* TWL$_{max} \geq 6.1$ m, $Q_{max} \geq 1000$ m$^3$ s$^{-1}$.

Figure 9 shows the spatial distribution of flooding for the above four cases for Scenario 3 (tide + max surge combined with river events and a $-3$ h time lag). The TWL-dominated event is shown in Fig. 9a, where water inundated the lower and middle estuary. The $Q$-dominated event simulated upstream flooding (Fig. 9b). The moderate compound event is shown in Fig. 9c, where the inundation pattern shows flooding mostly at the upstream region and part of the middle estuary. Finally, the extreme combined event is shown in Fig. 9d, where water inundated wide parts of the floodplains throughout the estuary. It can be seen that the flooded region of Fig. 9d is broadly the union of that in Fig. 9a and b.

The lateral extents of flooding, defined as the width of the inundated area in the direction perpendicular to the river channel, for Scenario 3 for cases (a–d) are presented in Fig. 10. In each case (a–d) three adjacent simulations are shown to depict some driver sensitivity. For the TWL-dominated case, the three simulations presented in Fig. 10a show extensive lateral inundation (15–60 m) simulated along the lower estuary floodplains (distance of up to 6 km from the estuary mouth), with limited inundation between 6–8 km, then extensive inundation further up-estuary (8–14 km) that was sensitive to $Q_{max}$ (in the range of 25–100 m$^3$ s$^{-1}$) and limited inundation beyond 14 km. For the three $Q$-dominated cases (Fig. 10b), extensive inundation (20–60 m) was simulated in the upper estuary (8–19 km) with minimal sensitivity between the three simulations. For the moderate compound event cases (Fig. 10c), simulated lateral inundation showed large sensitivity to forcing conditions, with up to 40 m variability between the three simulations at 10–14 km. The capacity of the estuary for floodwater storage is clearly sensitive in this region. Finally, for the extreme combined event cases (Fig. 10d), extensive lateral flooding (15–60 m) was simulated throughout the lower and upper estuary, except be-

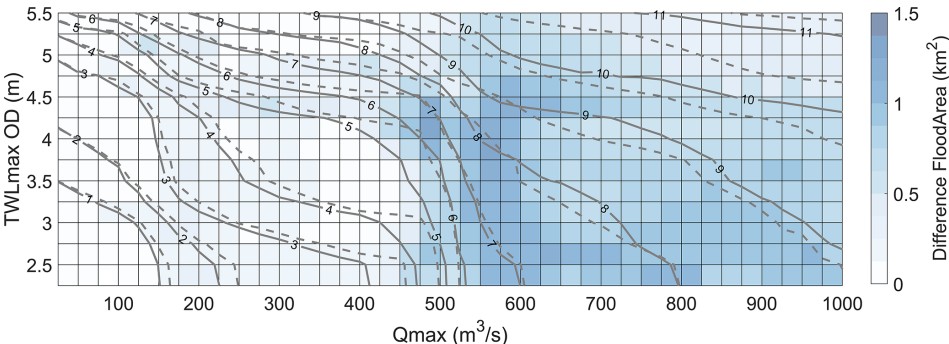

**Figure 8.** Coloured surface represents the absolute difference in modelled FloodArea between Scenario 1 (maximum surge with a 0 h time lag) and Scenario 3 (maximum surge with a −3 h time lag). The solid contours link common FloodArea magnitude for Scenario 3, whereas the dashed contours refer to Scenario 1 for comparison.

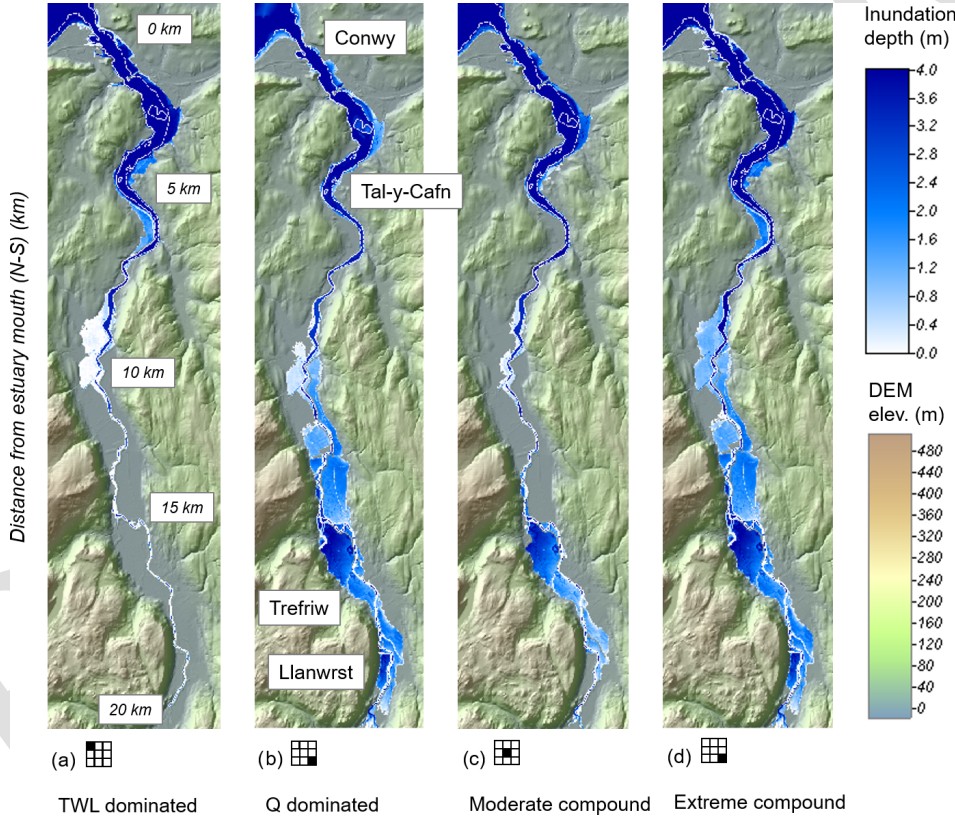

**Figure 9.** Scenario 3 (tide + max surge with river events and a −3 h time lag): simulated maximum flooded extent (blue shades) of the region of interest for cases of **(a)** TWL-dominated ($Q_1$TWL$_{13}$), **(b)** $Q$-dominated ($Q_{40}$TWL$_1$), **(c)** moderate compound ($Q_{20}$TWL$_7$), and **(d)** extreme combined ($Q_{40}$TWL$_{13}$). Corresponding FloodArea sizes are 5.6, 11.5, 8.9, and 6.6 km$^2$, respectively. The icons show the relative position of each case **(a–d)** in the TWL$_{max}$ : $Q_{max}$ parameter space (detailed in the Supplement). The dashed white lines delineate the shoreline in the no-flooding base case. The green–brown shading denotes dry land. CE1

tween 6–8 km where there was again limited flooding simulated. There was little sensitivity (< 1 m) between the three simulations shown.

## 3.5 Assigning probability to flood drivers

Figure 11 shows joint probabilities calculated from observed total water level at Llandudno and river discharge at Cwmlanerch, presented in the TWL$_{max}$ : $Q_{max}$ parameter space and overlaying the distribution of extreme events in the his-

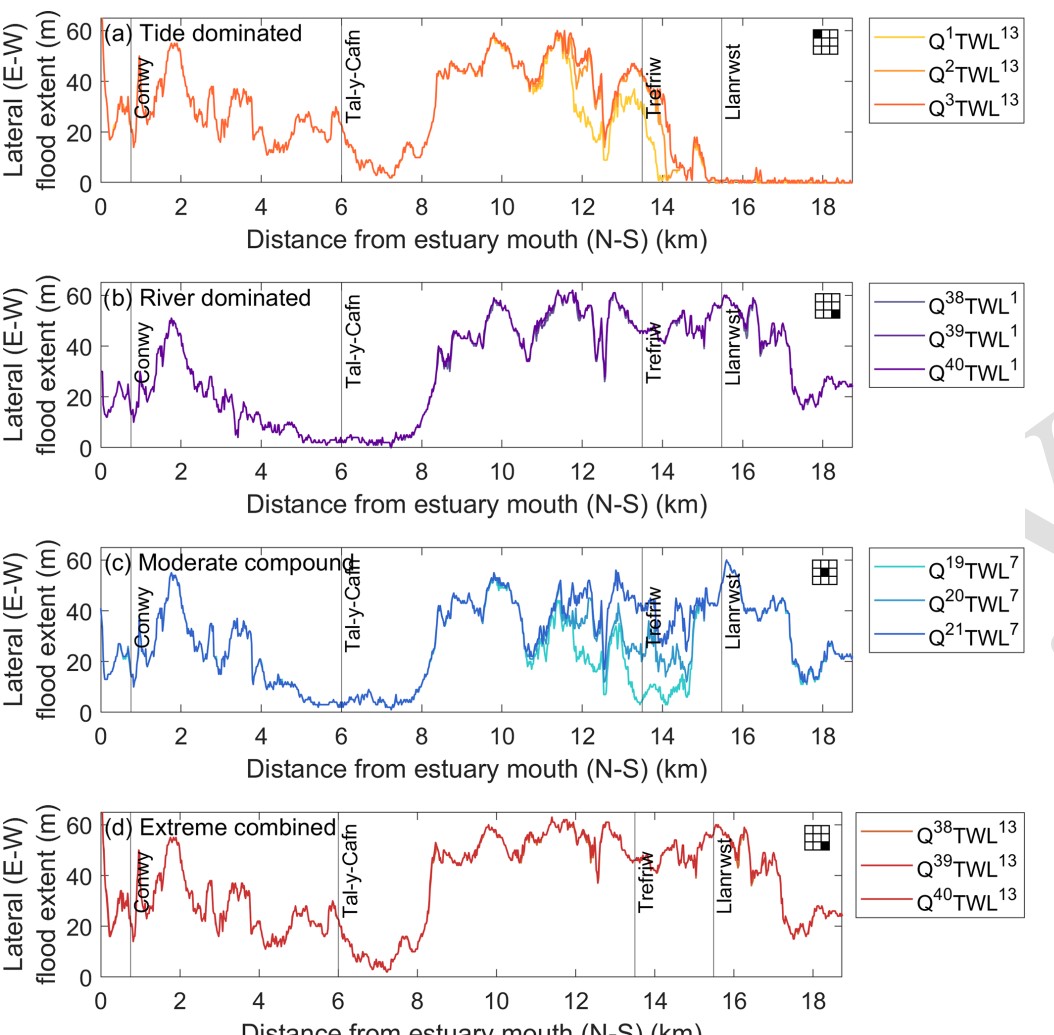

**Figure 10.** Scenario 3 (tide + max surge with river events and a −3 h time lag): distribution of lateral flooding along the Conwy estuary floodplain across the $TWL_{max} : Q_{max}$ parameter space for cases of **(a)** TWL-dominated ($Q_{1-3}TWL_{13}$), **(b)** $Q$-dominated ($Q_{38-40}TWL_1$), **(c)** moderate compound ($Q_{19-20}TWL_7$), and **(d)** extreme combination ($Q_{38-40}TWL_{13}$). Lateral flooding is measured in the east–west direction. Along-estuary distance is measured in the north–south direction (from the estuary mouth to upstream). For each case **(a–d)**, three simulations are presented (constant $TWL_{max}$ and varying $Q_{max}$; see also Fig. S3). The icons show the relative position of each case **(a–d)** in the $TWL_{max} : Q_{max}$ parameter space (detailed in the Supplement). CE2

toric record. Figure 11 represents a novel approach to interpreting joint probabilities in the context of historic storm events to better understand the relationship between drivers and impacts of flooding. The joint probabilities highlight the likelihoods and severities of the historic extreme compound events. There were seven historic events which have a probability of $< 0.01$, indicating less than one event in 100 years of this magnitude, six of which are recorded as causing flooding (yellow circles), whereas for one of these events no flooding was recorded (blue triangle). The no-flooding event was on 10 February 1997; $Q_{max}$ was 311 m³ s⁻¹, which peaked 1 h 30 min before $TWL_{max}$, recorded as 5.1 m, including a 0.48 m skew surge. Reports indicate this was a high-water-level event, associated with a 5-year sea level return period,

but these conditions did not cause flooding or no flooding was recorded (HR Wallingford, 2008). This method allows for return periods to be assigned to historic extreme events and recorded flood events and to estimate the likelihood and severity of potential future events. Figure 11 shows that the same joint probability can occur from a range of combinations of $Q_{max}$ and $TWL_{max}$ conditions. For instance, an event with a 0.2 exceedance probability (one event in 5 years) can occur on a TWL-dominated, $Q$-dominated, or moderate compound event.

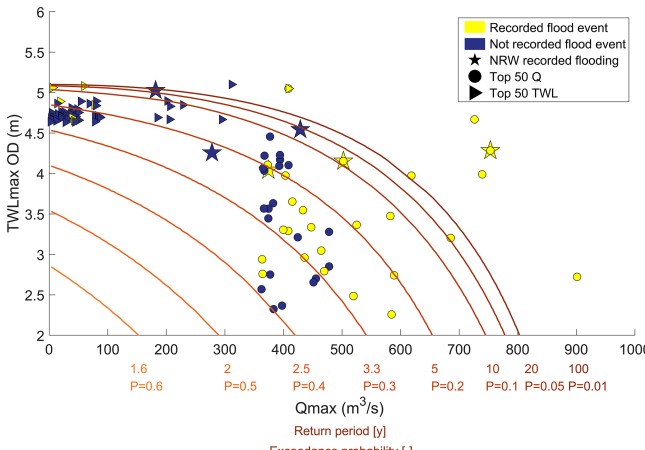

**Figure 11.** Joint probabilities for $TWL_{max}$ and $Q_{max}$ in the Conwy estuary, where $P$ is the exceedance probability, ranging from a high likelihood of co-occurrence ($P = 0.6$) to a low likelihood of co-occurrence ($P = 0.01$), overlaid by the distribution of extreme events (recorded and not-recorded flooding) in the historic record.

## 4  Discussion

This research has established site-specific driver thresholds for flooding in an estuary environment, using hydrodynamic modelling. The simulations have been verified and contextualised using documented records of flooding, together with data analysis and statistical analysis of instrumental gauge time series. With application to the Conwy estuary, N Wales, the hydrodynamic inundation model was applied to a series of idealised combined river and sea level compound events. We show that flooding is co-dependent on $TWL_{max}$, $Q_{max}$, and their relative time lag and that historic records of flooding can be used to set driver and flood extent thresholds that isolate minor and severe flooding. Below, we discuss the thresholds of flooding and the importance of accurate records of historic flooding events. We consider these thresholds may change under different driver behaviours and combinations and future climate conditions.

### 4.1  Thresholds for flooding

Since there are multiple drivers of flooding in estuaries, single-value driver thresholds cannot be used; e.g. for the Conwy estuary we show for the first time that flooding is co-dependent on $TWL_{max}$, $Q_{max}$, and their relative time lag. The simulated flooding presented in Sect. 3 shows the total inundation (FloodArea) across the estuary system and includes both minor or nuisance flooding up to severe flooding. Recorded flood events are isolated based on time lag and associated web-scraped tag(s) (cf. Sect. 2.3) and presented with FloodArea contours from Scenario 3 to identify if there is a simulated FloodArea threshold that matches the recorded flooding events (Fig. 12). The 2 or 3 km$^2$ contour lines can

be interpreted as a minimum FloodArea contour for recorded flooding in the Conwy. The coastal events (Fig. 12c) occur under a high sea level and across a range of river discharge combinations, indicating thresholds for flooding in the coastal zone should consider sea level as the dominant driver.

Whilst the FloodArea representation gives a good overall perspective of flooding dynamics, a different approach is needed to establish co-dependent driver thresholds for flooding at different locations within the estuary. For a chosen location, as a first step, a flood threshold (i.e. depth of inundation) has to be established. For instance, one might expect to assign a different flood threshold for an area of unused woodland than an agricultural field or a dwelling or road, based on socio-economic impact metrics (Cutter et al., 2013; Alfieri et al., 2016). Next, the inundation modelling shown in Sect. 3 can be used to predict whether flooding is likely to have occurred or not for the range of compound events within the parameter space and hence define the site-specific co-dependent driver thresholds. This is an approach often used for coastal infrastructure, including nuclear sites (e.g. ONR, 2021), but rarely extended to individual properties or land users. We have demonstrated this procedure below for four discrete locations within the Conwy estuary floodplains: (i) primary school, Conwy; (ii) farmland, middle estuary; (iii) section of railway, middle estuary; and (iv) dwelling, Llanrwst. We used Scenario 3 (tide + max surge combined with river events with a $-3$ h time lag) for this demonstration since this scenario predicted the most flooding. Figure 13 shows the co-dependent driver thresholds for each location (i–iv). Figure 13 shows TWL-dominated flooding in the lower estuary when sea level is $> 5.7$ m at the school and $> 4.9$ m at farmland and river-dominated flooding in the upper estuary at dwellings when river discharge is $> 750$ m$^3$ s$^{-1}$. This also aligns with what is shown in Fig. 10, and single-variable ($Q$ or TWL, respectively) flood probability analysis may be appropriate in these locations. Moderate compound flooding in the middle estuary shows flooding under a wider range of TWL and $Q$ combinations and shows that joint-probability analysis is necessary when both drivers influence flood magnitude.

#### 4.1.1  Flood dynamics related to driver magnitude and timing

We show that flood forecasts need to be sensitive to both fluvial and sea level drivers of flooding in the Conwy estuary, N Wales, particularly under medium levels (45th–60th percentiles) of river discharge and total water level. Flood hazard assessments must consider a bivariate approach to both river discharge and sea levels across an estuary; otherwise univariate approaches will not appropriately characterise the hazard and will underestimate compounding effects (Moftakhari et al., 2017). Combined river and sea level simulations show that when the drivers are extreme (e.g. $> 85$th percentile),

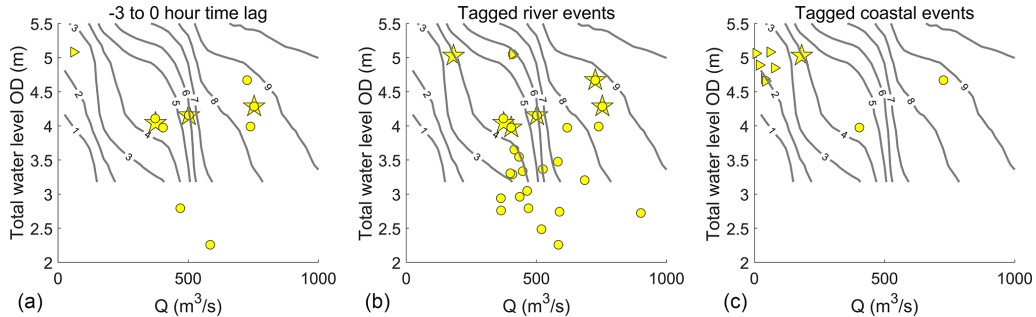

**Figure 12.** Recorded flood events with **(a)** a time lag between 0 and $-3$ h, **(b)** tagged [1] river events, and **(c)** tagged [3 4] coastal events (web-scraped keywords), all presented with FloodArea contours from Scenario 3. Web-scraped keywords are explained in Table 1. CE3

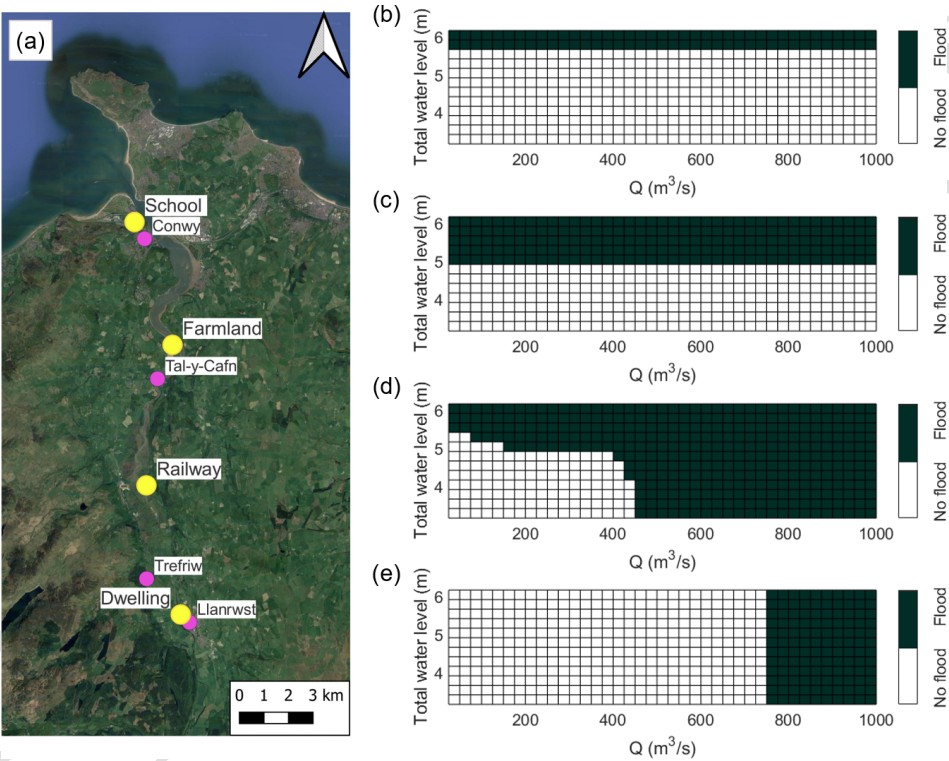

**Figure 13.** Site-specific flood thresholds to show the conditions that cause flooding to occur or not within **(a)** the Conwy estuary using model outputs from Scenario 3 at **(b)** a primary school in the lower estuary, **(c)** farmland in the lower estuary, **(d)** a railway in the middle estuary, and **(e)** a dwelling in the upper estuary. **(a)** Basemap © OpenStreetMap contributors 2023. Distributed under the Open Data Commons Open Database License (ODbL) v1.0.

they act equally and consistently produce the highest magnitudes of flood inundation irrespective of their relative timing. The volume of riverine freshwater is the dominant driver contributing to high water levels in the estuary. This could be evidence of the backwater effect, where high river discharge can push back low levels of tidal water, resulting in a temporary increase in water levels within the estuary (Ikeuchi et al., 2015; Feng et al., 2022).

Results show that flood forecasts need to be particularly accurate for the Conwy estuary when the river discharge is between 450–550 $\mathrm{m^3\,s^{-1}}$, which represents moderate conditions. We show that within this range of discharge there is considerable variability in flood inundation across a range of sea level magnitudes, which is also sensitive to the timing of $Q_{\mathrm{max}}$ relative to $\mathrm{TWL_{max}}$. This critical range of discharge values, between 450–550 $\mathrm{m^3\,s^{-1}}$, could be related to the holding capacity of the estuary as there may be storage volume for flood water below these magnitudes of discharge. This critical range of discharge values also represents a threshold for a change in the behaviour of the drivers. Analysis of FloodArea contour shapes/gradients superimposed on historic flood inundation records shows that compound ef-

fects are most significant under medium levels of river discharge and sea level. Below these medium levels, one or the other driver is more dominant. Above this level, both drivers are equally dominant in their contribution to flooding. These insights show that both drivers must be considered dependent and interacting in flood forecasts to ensure that compound flood effects are captured and planned for.

An analytical model has been used in an idealised, mesotidal estuary to show that there is always a point where river discharge effects on water level outweigh tide–surge effects (Familkhalili et al., 2022). Non-linear effects and interactions between sea level and river discharge can influence compound effects, including tidal damping and tidal blocking, and can influence the location at which river flow effects are larger than marine effects, or vice versa (Cai, 2014; Hoitink and Jay, 2016; Xiao et al., 2021). The magnitudes at which river discharge and sea level will cause compound effects to amplify flood inundation will vary between estuaries. These effects may not occur in some estuaries and may be more extreme in others (Harrison et al., 2021). It is likely that a range of factors will control this including tidal range; substrate type and bed friction; coastline aspect; estuary geometry and size; catchment size, type, and geology; river network; river transmission times; prevailing weather conditions; antecedent weather; and local climate (Familkhalili et al., 2022). The parameter space could be developed by considering additional hydrograph time lags and exploring the timing of the surge relative to tidal high water which could influence the magnitude and volume of the total water level (Lyddon et al., 2018; Khanam et al., 2021). The lag time is currently presented as between $Q_{max}$ and $TWL_{max}$; however there could be asymmetries within the estuary that prevent tidal slack water from occurring at $TWL_{max}$. The $Q_{max}$ lag relative to slack tide (e.g. turning from flood to ebb) could be explored; however significant 3D lateral flows in the Conwy estuary (e.g. Robins et al., 2012; Howlett et al., 2015) would mean that identifying the location and timing of slack water would require a 3D baroclinic model. These additional parameters could alter the position, shape, or angle of threshold contours or understanding of flood dynamics. A better understanding of estuarine thresholds can enhance how managers and engineers plan coastal protection strategies, including where to place defences, infrastructure, and buildings.

## 4.2   Documented records of flooding

Historical records of flooding in the Conwy estuary are incomplete, with few flooding events from before 2004 documented and available online. More recent flooding events have only been recorded online unsystematically and are contingent on the severity of the impact, suggesting that smaller flooding events or flooding away from people and infrastructure have potentially been undocumented. Additionally, documented flooding events tend to focus on the impacts rather than the drivers that caused the hazard. This study adds to the historical catalogue of flooding in the Conwy estuary by collating all available documented events into one space together with the driving river flow and sea level conditions and their relative timings. We believe that similar circumstances of incomplete historical records of estuary flooding are widespread nationally, and indeed there is limited knowledge of how estuary flooding has varied geographically. National UK chronologies of flash flooding (Archer et al., 2019) and coastal flooding (Haigh et al., 2015) have been compiled, but such records do not exist for estuaries.

Documenting compound flood events aids in understanding and analysing the drivers, interactions, and impacts of the hazards (Haigh et al., 2015, 2017); validating numerical and statistical techniques; and calculating optimal thresholds. Recording historic information on river flows/levels, sea levels, other sources such as pluvial and groundwater flows, and subsequent flooded areas helps to identify high-risk areas and areas where appropriate measures to reduce future flood risk may be required. This prior knowledge combined with current information on where and when certain combinations of extreme conditions are forecast can aid in incident response for flood agencies and emergency services and help local authorities identify what resources are needed in the short and longer term following flooding. Comprehensive historic flooding records can provide an opportunity to assess the effectiveness of existing flood management policies and flood control measures, such as flood walls or drainage systems, that need improvement. This knowledge can guide future engineering designs for a range of coastal development, ensuring the construction of more resilient and adaptive infrastructure that can better withstand flood events. Documenting flood events can also build a database of information to help raise public awareness of and resilience to flood hazards. Photographs, videos, and written accounts of past events can evoke an emotional response to prompt individuals and communities to engage with future flood preparedness and evacuation plans (Fekete et al., 2021; Wolff, 2021). These data could also be extended to include storm tracks, storm footprints, rainfall intensity, groundwater levels, and catchment saturation to build a greater understanding of the meteorological conditions that can contribute to compound flooding events (Zong and Tooley, 2003). Social media data, including geolocated tweets, have been used to identify the remarkability of events and highlight major cities, including Miami, New York, and Boston, that are vulnerable to flooding (Moore and Obradovich, 2020). Qualitative hazard data from archived and digitised newspaper articles have been extracted to identify the geographic location, date, triggers, and damages of estuarine floods (Rilo et al., 2022) and validate flood models (Yagoub et al., 2020).

The combined approach to identify driver thresholds for compound flooding presented here, as well as additional parameters suggested to develop the approach, relies on availability and access to sufficient instrumental data at the appropriate temporal resolution and topographical and bathy-

metric data at appropriate spatial resolution. The UK sea levels, river discharges, and topography are recorded, archived, and accessed via national government and research agencies (e.g. British Oceanographic Data Centre, National River Flow Archive, Centre for Environment, Fisheries and Aquaculture Science, and Channel Coastal Observatory). However, nearly 50 % of the world's coastal waters remain unsurveyed (IHO C-55, 2021), and 290 tide gauges that form the Global Sea Level Observing System (GLOSS; Merrifield et al., 2009) are unevenly distributed across the globe and do not account for local, vertical land movements. The approach described here could supplement existing observation systems with new technologies to improve records of coastal processes (Marcos et al., 2019), at local scales including X-band radar-derived intertidal bathymetries (Bell et al., 2015; Bird et al., 2017) and X-band radar-derived tide and surge (Costa et al., 2022) and regional scales including satellite-derived bathymetry (Cesbron et al., 2021; Hasan and Matin, 2022) and satellite altimetry (Cipollini et al., 2016), which measures the sea level from space with sufficiently dense global coverage. Global model projections of storm surge and tide can be downscaled and applied to inform assessment of coastal flood impacts (Muis et al., 2023). Temporal and spatial gaps also occur in the global river discharge observing network, and hydrometric data are not available in real time (Lavers et al., 2019; Harrigan et al., 2020). Research has focused on coupling surface and sub-surface runoff models, hydrologic models, and land surface models, which are forced with global atmospheric reanalysis (e.g. ECMWF's ERA5) to produce river discharge reanalysis (Harrigan et al., 2020). Combining observation and downscaled modelled data to explore thresholds for estuarine flooding is one approach to apply this methodology worldwide.

Improving the resilience and preparedness of communities to flood hazard is a UK priority policy, as outlined in the Defra (Department for Environment Food & Rural Affairs) policy statement on flooding (2021), highlighting the need for integrated approaches to flood hazard management. Instrumental data can be used in conjunction with earth observation records, including remote sensing and satellite imagery, of flooding to build more comprehensive databases of past records of estuarine flooding and be supported with numerical modelling studies to help identify thresholds for flooding (Heimhuber et al., 2021; Costa et al., 2023).

## 4.3 Future changes in flooding

Extreme sea levels for the Conwy, comprising large spring tides and large skew surges, could reach ∼ 6 m (OD) and were simulated here in the upper rows of the scenario parameter space. These levels have not yet been seen in the Conwy but could happen presently. The FloodArea contours are close together in this section of the parameter space and show that relatively small increases in sea level and/or river flows lead to large increases in flood extent. This section of the parameter space is likely to become more relevant in the coming decades, as a result of sea level rise and projected increases in the magnitudes of peak river flow events under future climate conditions. Sea level rise and geomorphic changes will lead to a new baseline for flooding and new driver thresholds and interactions. Many studies have started to consider the impact of climate change on compound estuary flooding (Robins et al., 2016; Ghanbari et al., 2021). Outputs of climate models were analysed to show that changes in sea level and precipitation can substantially increase the likelihood of a compound event, where a 100-year event could become a 3-year event by 2100 (Peter Sheng et al., 2022). Model simulations of synthetic storms of combined tropical cyclones and sea level rise in the Cape Fear estuary, North Carolina, have shown that future climatology will increase a 100-year flood extent by 27 % (Gori and Lin, 2022). In addition to future changes in drivers of compound events, it is possible that changes in storm tracks will influence the clustering and timing of events (Haigh et al., 2016; Eichentopf et al., 2019) and changes in land use could influence groundwater saturation, baseflow, and overall floodwater storage and drainage capacity of the system (Rahimi et al., 2020). However, uncertainties in future UK projections of river discharge and sea level must be accounted for when considering compound flood effects (Lane et al., 2022). It is beyond the scope of this research to explore the influence of future climate changes on thresholds but could be explored by running simulations with different groundwater saturation, clustered events, and higher sea level or river discharge. A better understanding of how compound events and thresholds will change in the future is also crucial for developing adaptive strategies for high-impact events (Zscheischler et al., 2018), and climate projections of changing sea level, storm surge, river discharge, and storm tracks should be considered in model scenarios.

## 5 Conclusions

The urbanisation and industrialisation of estuaries have increased the vulnerability of communities to extreme events, such as flooding from high sea levels and river discharge. The impacts of these events are further amplified when extreme sea and river events occur simultaneously. Flooding occurs when coastal or fluvial conditions exceed critical thresholds such as flood defence heights, so there is a need to identify the driving land and sea conditions under which these thresholds are exceeded and the type of flooding that ensues. This research developed a novel framework that utilised a combination of historic estuary flooding records, instrumental monitoring data, numerical modelling, and probabilistic analyses to identify driver thresholds for compound flooding, for an estuary that is especially vulnerable to compound flooding events (Conwy, N Wales, UK).

The simulations predict how the total estuary flooding extent responds to the magnitude of river discharge, tide, and surge magnitude and the timing of peak river discharge relative to tidal high water. Most flooding occurs when one or both sea level and river discharge drivers are extreme (e.g. > 85th percentiles) but with amplified (compounding) flooding under relatively moderate circumstances (e.g. 60th–70th and 30th–50th percentiles) and in specific regions of the estuary (middle estuary). Flooding is sensitive to a change in the timing of peak river discharge relative to tidal high water, with a −3 h time lag (peak river discharge 3 h before high water and coinciding with a rising tide that "traps in" the freshwater), causing 7.7 % more flooding across the parameter space than with a 0 h time lag. There is spatial variability in flooding that is dependent on the combination and magnitude of the drivers. We show in detail the simulated extent of flooding in the lower estuary under extreme sea level conditions and in the upper estuary from extreme river flow conditions – as well as the spatially intricate nature of flooding throughout the estuary under combined moderate and extreme ("worst-case") sea level and river flows.

The research highlights that the recorded flooding extents held by national agencies are incomplete. This database is important to build knowledge on past flooding episodes (e.g. when and where has flooded and under what conditions), undertake further analyses such as temporal trends in flooding, and develop accurate and timely flood warnings. The historic flooding record for the Conwy was supplemented with information obtained from online sources available for 2004–2022 and set within the context of the most extreme 100 compound events during the period 1980–2022. An estuary inundation model was then used to "fill" the parameter space of possible compound events (1560 separate simulations). This combined approach of modelling referenced to historic flooding events allowed us to identify a range of thresholds for flooding.

The results highlight under which conditions flooding is predicted to occur, or not, throughout the estuary and identify driver thresholds for flooding that are relevant to historic recorded flooding, steep increases in flooding (sensitive tipping points), and location-specific/impact-specific flooding. The method can be used to enhance our understanding of estuarine flooding dynamics and improve flood risk assessments – it can be applied to other estuaries worldwide where there are paired coastal and fluvial monitoring/model data, and the methodology can be developed to include additional drivers and changes in the timing of behaviour of the drivers surges under different climate/management conditions.

*Code and data availability.* All code and raw data can be provided by the corresponding authors upon request.

*Supplement.* The supplement related to this article is available online at: https://doi.org/10.5194/nhess-24-1-2024-supplement.

*Author contributions.* CL, NC, GV, PR, AB, and TC formulated the research and developed the methodology; GV and TC developed, calibrated, and validated the model setup; NC ran the model and managed model outputs; AO, SM, and MR contributed to the data analysis; CL, NC, and PR analysed and visualised the results; CL wrote the manuscript draft; PR, NC, GV, AB, TC, and AO contributed to, reviewed, and edited the manuscript.

*Competing interests.* The contact author has declared that none of the authors has any competing interests.

ther geographical representation in this paper. While Copernicus Publications makes every effort to include appropriate place names, the final responsibility lies with the authors.

*Acknowledgements.* The authors wish to acknowledge the NERC UK Climate Resilience Programme project SEARCH (grant no. NE/V004239/1), in partnership with Jason Lowe, Rachel Perks, Jonathan Tinker, and Jennifer Pirret at the Met Office; Mark Pugh at Natural Resources Wales; Sue Manson and Harriet Orr at the Environment Agency; and Fiona McLay at the Scottish Environment Protection Agency. The authors also acknowledge Cllr Aaron Wynne at Conwy County Borough Council and John Owen and Robert Meyer, who are residents and landowners in the Conwy floodplains, for their knowledge on flooding in the region.

*Financial support.* This research has been supported by the Natural Environment Research Council (grant no. NE/V004239/1).

*Review statement.* This paper was edited by Kai Schröter and reviewed by two anonymous referees.

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

TS5    Please confirm year.