# Peer review of "Thresholds for estuarine compound flooding using a combined hydrodynamic-statistical modelling approach"

_EGUsphere, 2023_

## Referee Comment (RC3)

The manuscript presents an interesting framework for setting thresholds for estuarine compound flooding using combined hydrodynamic statistical techniques. The approach is applied to Conwy Estuary, North Wales, a particularly vulnerable area to compound flooding. It represents a current thematic area and can be particularly useful in improving compound flooding assessment in estuaries worldwide. The study's main objective is to identify the coastal and fluvial conditions that lead to flooding in an estuarine system.

The manuscript is well-written (English level) and presents high-quality scientific content with interesting results and discussions. No doubt, a lot of work has been done, and the research is highly relevant. However, the connection between chapters (structure) needs improvement. In addition, the authors make some general assumptions throughout the text about specific terms that can confuse the reader, especially if he/she is not familiar with the United Kingdom or the field of coastal/estuarine flooding (especially in the Introduction and Methods Section). Further explanations need to be given in these specific points to improve the reader's comprehension for a broad scientific audience.

Therefore, I do not recommend publishing this manuscript before major revisions.

My main critics are the following:

1. In Section 1 (Introduction), the importance of the topic is well developed, although some key elements need some clarification. For example, the authors talk about thresholds being fundamental to assessing compound flooding; however, they do not explain what a threshold is or why they are important from a technical point of view.

2. In section 2 (Methods), much information is given; however, some details and the connection between the subsections must be improved. For instance, why the authors used the recorded data (subsections 2.2 and 2.3) is not well explained. What is the objective of collecting that information? Is it for validation of the hydrodynamic simulations? A clear statement (short paragraph) should be written right at the beginning of the subsection, as it was done for subsections 2.4 and 2.5. Subsection 2.6 needs the same explanation at the beginning of the paragraph.

3. The authors use specific terms not defined anywhere in the text—for example, skew-surge, storm surge, total water level, flood event. Although familiar to the coastal flooding research public, they must be defined the first time they are mentioned in the

manuscript.

4. In sections 2.1 and 2.2, several flooding events are described; however, a general description of storm surge climate and tidal regime in Conwy Estuary is missing.

5. It is hard to follow section 2.2. and the reasoning behind the selection of flooding events. The authors mentioned that only six flood events were recorded in TWL and river discharge; however, further in the text, they mentioned the top 50 TWL and Qmax events. The authors need to make this explanation more clear. In addition, the authors also need to explain better what was considered a flood event. This general information should be shown at the beginning of the section, followed by a detailed description of single events already presented in the text.

6. Section 2.4. Is the model solving the equations of movement in 2D or 3D? Please clarify it. In addition, if it is solved in 2D, please justify why it is not solved in 3D and vice-versa.

7. In subsection 2.5, a general description should be added to the first paragraph of the subsection, explaining the number of scenarios, which are the drivers tested (total water level, storm surge, river discharge, time lag) and the simulation period (72 hours). I know this information is given through the section; however, it is scattered and confuses the reading.

8. Section 2.8 does not clearly explain why joint probabilities were used. Why is it important? This is partially explained in the introduction and should be remembered here. Similarly, a further explanation of why Copulas is applied from a technical point of view should be given. For instance, are Copulas used to generate synthetic samples of extreme sea levels and river discharges, thus making their respective probability distribution more robust to apply joint probability methods? A general explanation (short and from a practical point of view) should be given about how statistical marginal distributions, copulas, and Bayesian methods are linked to each other.

9. In Section 2.8, there are references to several tables from another paper. I do not believe this is a good way to mention it. If applied, this content should be added to supplementary data or referred to the work without the table number.

10. Section 4 (Discussion) lacks a clear first paragraph, generally stating the manuscript's main findings. The first paragraph is too vague and similar to the

conclusion section's first paragraph. I suggest making a general statement on the main findings and then detailing each in the following subsections.

11. In Section 4.1, I do not understand why the documented flooding records are discussed if they are not even in the section Results (they are shown in Methods). I suggest shortening it to conclusions or moving it to the end of subsection 4.2, "Thresholds for flooding", where you could link the importance of historical records of flooding events to validate numerical/statistical techniques and calculate optimal thresholds (for instance).

12. Section 4 shows several interesting discussion points; however, the authors did not discuss much about the novelty of the method. How innovative is the approach applied here? For instance, Section 4.2 shows a good discussion about setting thresholds from an end-user point of view (flooding mitigation planning); however, the authors do not discuss the hydrodynamic-statistical approach itself.

13. To my understanding, the approach presented here relies on having sufficient data (recorded flooding, water level and river discharge observations, good quality topo-bathymetry data), which is not the reality of several regions globally (perhaps also in the UK). I see the authors discussed that in section 4.2; however, the discussion is focused on the UK. I suggest adding a short discussion about the quality of the forecast and data availability on regional (e.g., Europe) and global scales.

14. In section 5 (Conclusions), the authors wrote a good introductory first paragraph; however, just after, they talk about historical floods, which is not the paper's main goal (I suggest deleting this paragraph or shortening and moving it to the end of the section). I would have expected that they answered the main objective straight after the introductory paragraph (to identify the coastal and fluvial conditions that lead to flooding in an estuarine system).

SPECIFIC COMMENTS

15. Lines 6 – 11. Please replace "UK" with "United Kingdom".

16. Line 14. Please replace "UK" with "United Kingdom".

17. Line 19. Please replace "N-Wales" with "North Wales".

18. Lines 20–22. It is not clear what was amplified. What does sensitivity 7% mean?

19. Line 30. Replace "–" (en-dash) with "—" (em-dash).

20. Line 34. I understand the UK refers to the United Kingdom; however, the authors should define all the acronyms. This acronym has not been previously defined in the text. Suggestion: "United Kingdom (UK)". Then, after that, you could only use the UK.

21. Line 80. "Modelling statistical and probabilistic methods," wouldn't they be the same?

22. Line 89. "N-Wales". Please homogenise the use of "North Wales". If "N-Wales" is used, then it needs to be defined, for instance: "North Wales (N-Wales)....". Then, use only N-Wales after that.

23. Line 110. In "November 1980 - February 2023", a dash is inappropriate. En-dash "–" should be used for ranges of dates and numbers. Please replace throughout the document whenever applicable.

24. Lines 126–133. The terms "total water level (TWLmax)", "predicted tide level", "skew surge", and "storm surge" are not defined. For instance, does the "total water level" include the sea-level anomaly/trend? If so, the sea level trend could interfere with your results. This should be well clarified.

25. Line 130. Please define the acronym "NRW" and homogenise the use of Natural Resource Wales. Use "Natural Resource Wales" or NRW throughout the document (e.g., line 138).

26. Figure 1. The labels of the x-axis in panels (e) and (f) do not follow the same pattern. Please homogenise them.

27. Line 150. What is "event hydrographs"? Please define it.

28. Line 177. Please change "however," to "; however, ..." or ". However, ..."

29. Line 188. I am not used to the term "Web scraping approaches". Is that a proper term to be used in a scientific paper? Maybe change it to "Web searching"?

30. Table 1. It seems to be cut at the bottom of the preprint. Please check if this is indeed the case. Bellow a screenshot:

| Label | Code |
|-------|------|
| 0 | None |
| 1 | River discharge |
| 2 | Storm surge |
| 3 | High tide |
| 4 | Storminess |

31. Line 201. In the sentence "… with yellow dots indicating there is evidence of flooding and blue dots indicating there is no evidence of flooding." How can it not have evidence of flooding and be on the internet? I couldn't understand it.

32. Lines 209–210. "…. leave uncertainty in where to set driver thresholds and patterns for flooding, especially for less extreme Qmax and TWLmax that led to compound flooding.". The concept of threshold (quantiles, peaks-over-threshold, block maxima) and event definition (How long an event was considered to last? Was used any declustering schemes?) needs to be clearly described in the introduction or previously in the methods section.

33. Line 213. I do not fully understand what the authors considered a top 50 Qmax and TWLmax. The authors mentioned that only a few recorded flooding were identified (6 of NWR and 20+ in web search). However, in Figures 2 and 3, the authors show more events than that. Top 50 events mean that you have selected the top 50 events, or are you taking the events above the 50% percentile? Please make it more clear.

34. Line 224–226. The main sources for DTM, bathymetry, and flood defence locations should be mentioned in the main text.

35. Line 245–248. How exactly did the authors gradually adjust the channel bed elevations? Manually editing the bathymetry? The Neal et al. (2022) work should be better described. One or two short sentences should be enough. Also, what is a "stepwise manner"?

36. Line 249. Why did the authors use two scores (RMSE and Kling-Gupta Efficiency)? Is there any advantage to using that? Formulas should be added to the supplementary material.

37. Line 253. "in the upper estuary". Please add the names of the stations in parentheses.

38. Line 254: "tributaries". Please explain.

39. Line 270. "The M2 tidal constituent has an amplitude of 2.71 m and was used to produce a constant sinusoidal curve for 72 hours". Why only the M2? Are the shallow-water harmonics not important in this estuary? Why 72 hours?

40. Line 272. "scale factor of 25 cm…. thus creating 13 water level time series". Sorry, I could not understand the reasoning. Between 1.82 m (high neap tide) and 3.6m (high spring tide) and a scale factor of 25 cm (adding 25 cm to 25 cm), I could only count seven water level time series and not 13. Please clarify it.

41. Line 276. What is a representative surge shape?

42. Line 297. What does "spin-up" time mean?

43. Line 306. Where exactly does the 40 discharge time series come from? Please clarify.

44. Figure 5. Is the y-axis label of the panel (c) correct ("Number of events")? Shouldn't It be "Total water level OD (m)"?

45. Line 343. Please replace "(ROI, see Figure 1a)," for "(ROI), see Figure 1a, .."

46. Lines 348–358. Please consider joining this paragraph with the previous one.

47. Line 357. Please clarify why 520-simulation parameter space Scenarios 1–3. I understood there were 1560 simulations.

48. Lines 360–366. Please refer to what section will show and discuss these results. It is confusing to the reader to know if you are talking about the simulation scenarios

previously described or the spatial analysis of the flooding area. In addition, what does lateral flood extent mean? Please clarify and define it.

49. Line 368. Why do the authors want to use joint probabilities? At which part of the method the authors are applying this? Is there anything to do with hydrodynamic modelling? Please make it more clear.

50. Line 371. "to the data", which data?

51. Line 378. "Table 6 of Moradian et al. (2023)". Please see comment 9. Apply the comment throughout the text.

52. Line 380. Why so many metrics? Please justify.

53. Line 403. "dependence measures". I suggest changing this term to "dependence metrics" or "correlation coefficients" and then removing "Correlation Coefficient" after the name of each coefficient you mentioned just after. Also, the authors need to explain further why dependence metrics are important. What is it used for?

54. Line 407. The authors need to explain better why they want to use Bayesian methods and how they did it.

55. Line 415. "is the probability of A being true and". The subject of the sentence is missing.

56. Lines 423–432. Please join these two paragraphs. The second one gets confusing when not directly linked to Figure 6.

57. Lines 446–448. Couldn't it be a question of scale in the graphics? Is the figure showing a normalised plot? If not, you compare different units (m) and (m3/s). I would plot a normalised plot to double-check this question. Please correct it if applicable.

58. Figures 6, 7, and 8. What is OD in the y-axis label? Is it the vertical datum? It should be explained in the figure caption.

59. Line 491. Sometimes, the authors use FloodArea in italics, and sometimes they do not use it. Please homogenise it throughout the manuscript.

60. Line 519. Please replace "9d" with "Figure 9d".

61. Line 524. Please explain the terms Q1Twl13, Q40 TWL1, Q20TWL7 and Q40TWL13. I could not understand why the authors used them.

62. Line 526. What is "TWLmax: Qmax parameter space"? Is that figure 6? Please refer to the figure or provide further explanation.

63. Line 530. What does "lateral extent of flooding" mean?

64. Figure 10. Same as Figure 9. A better explanation of the numbers following Q and TWL is needed. Also, the use of the icon is not clear.

65. Lines 569–575. Please see comment n° 10.

66. Line 578. "piecemeal fashion". Is that a scientific term? Please replace it.

67. Lines 609–611. You can also mention that earth observation records can supplement estuarine topo-bathymetry and geometry data for multiple purposes, including hydrodynamic modelling. Reference suggestions:

*Valentin Heimhuber, Kilian Vos, Wanru Fu, William Glamore, InletTracker: An open-source Python toolkit for historic and near real-time monitoring of coastal inlets from Landsat and Sentinel-2, Geomorphology, Volume 389, 2021. https://doi.org/10.1016/j.geomorph.2021.107830.*

And

*Costa, W. L. L., Bryan, K. R., and Coco, G.: Modelling extreme water levels using intertidal topography and bathymetry derived from multispectral satellite images, Nat. Hazards Earth Syst. Sci., 23, 3125–3146, https://doi.org/10.5194/nhess-23-3125-2023, 2023.*

68. Section 4.2 is confusing. It seems the authors are introducing new results instead of discussing the current results. I could understand the relevance of the discussion; however, I suggest the authors re-write parts of the section to clarify that new results are not being shown.

69. Line 616. What is "web scraped tag(s)"? Was it explained anywhere in the manuscript?

70. Line 619. I do not follow the statement, "The coastal events (Figure 12c) occur across a range of river discharge combinations, and thresholds 620 may not need to consider this driver". Figures 12 a and b show that flooding events (time lag and river) occur in a similar range of river discharge to coastal events. Please make it more clear.

71. Figure 12. Is Sea Level ODN the same as Total Water Level OD? Please explain why the axis labels are different. The panel indication (a), (b), and (c) are not shown.

72. Line 653. Please clarify which ranges are considered extreme in parenthesis.

73. Lines 654–656. "The volume of riverine freshwater is the dominant driver contributing to high water levels in the estuary. This could be evidence of the backwater effect, where high river discharge can push back low levels of tidal water, resulting in a temporary increase in water levels within the estuary". Please provide some references that corroborate it.

74. Lines 658–659. Please re-order the sentence "It is when the river discharge is between 450-550 m3/s in the Conwy Estuary that flood forecasts need to be particularly accurate. " to "Results shown that flood forecasts need to be particularly accurate for Conwy Estuary when the river discharge is between 450-550 m3/s". In addition, please say in parentheses if this range of values is mild or extreme.

75. Line 679. In the sentence: "The parameter space could be developed by considering additional hydrograph time lags and exploring the timing of the surge relative to tidal high water, which could influence the magnitude and volume of the total water level (Lyddon et al., 2018; Khanam et al., 2021)." I suggest two references:

*Costa W, Bryan KR, Stephens SA and Coco G (2023) A regional analysis of tide-surge interactions during extreme water levels in complex coastal systems of Aotearoa New Zealand. Front. Mar. Sci. 10:1170756. doi: 10.3389/fmars.2023.1170756*

And

*Arns, A., Wahl, T., Wolff, C., Vafeidis, A. T., Haigh, I. D., Woodworth, P., et al. (2020). Non-linear interaction modulates global extreme sea levels, coastal flood exposure,*

*and impacts. Nat. Commun. 11, 1–9. doi: 10.1038/s41467-020-15752-5*

76. Line 688. "Sea-level rise and geomorphic changes will lead to a new baseline for flooding and new driver-thresholds and interactions ". Reference suggestion:

*"Khojasteh, D., Glamore, W., Heimhuber, V., and Felder, S. (2021). Sea level rise impacts Estuar. dynamics: A review. Sci. Total Environ. 780, 146470. doi: 10.1016/j.scitotenv.2021.146470"*

77. Lines 713–716. "The research highlighted the incomplete nature of recorded flooding extents held by national agencies, which are important to build a database of past episodes of flooding (e.g., when and where has flooded, and under what conditions) and undertake further analyses such as temporal trends in flooding. Such a database is crucial for developing accurate and timely flood warnings. ". This passage is a bit unclear; maybe change it to

"The research highlights that the recorded flooding extents held by national agencies are incomplete. This database is important to build knowledge on past flooding episodes (e.g., when and where has flooded, and under what conditions), undertake further analyses such as temporal trends in flooding, and develop accurate and timely flood warnings."

78. Section 5. It is confusing that historic flooding records are included in the Conclusion section but not in Results. Instead, they are described in Methods. I suggest removing the historic events from the conclusion or moving the historic flooding records from methods to results.

79. Section 5. See comment n° 14. Suggestion: the third and fourth paragraphs should be joined together and placed as the second paragraph. The paragraph in lines 713–720 should be the third or last.

---

## Author Response (AR1)

Thank you for the opportunity to submit a revised version of our manuscript, entitled *'Thresholds for estuarine compound flooding using a combined hydrodynamic-statistical modelling approach'.* The authors found the editor and reviewer's comments to be rigorous and constructive, and as such, we have taken care to respond to all suggestions and concerns. For clarity, line numbers refer to the tracked changes manuscript. The result is a majorly changed, significantly restructured, and much improved manuscript.

**Remarks from the preceding review file validation**

With the next file upload request, please check your "Author contribution": IO is not among the author list. (do you mean AO?).

Thank you, this has been corrected to AO.

Please also remove the example figure on page 42.

This has been removed.

I also noticed that your figures 1 and 13 contain maps and aerials. To clarify whether a copyright statement or a credit must be given in the map itself or in the caption, we differentiate between (a) maps entirely created by you, (b) maps created by you but based on layers reused from other originators, or (c) maps simply reused from other originators. An example for (a) is a digital elevation model (DEM) purely based on measurement points collected by you and derived by using a software product. If you use an existing map layer from another originator as a basis for significantly enriching the map with your own content, this would be an example for case (b). Case (c) could be a pure reproduction of Google Maps where your own contribution is rather small (e.g. a city map where you only added a few marks for your study locations). If the map was entirely created by you (case a), there is no need to change the caption or map. Please simply inform us. To the contrary, if your map follows cases (b) or (c), please let us know whether the map is distributed under public domain. If yes, please do not include a copyright statement (copyright is waived) but consider adding a credit to the map or caption. However, if your map follows cases (b) or (c) and is not distributed under public domain, please include at least a credit or even a copyright statement (e.g. © Google Maps), if this is required by the map provider, in the map itself or in the caption.

The following statement has been added to the caption for Figure 1 and 13:

L159: Figure 1a-c Basemap © OpenStreetMap 2023

Line 677: Figure 13a Basemap © OpenStreetMap 2023

**Response to review 1**

This manuscript by Lyddon et al. clearly sets out an approach to identify tidal and river flood hazard thresholds for an estuary to determine the dominant or compound drivers of flooding. This understanding is important for local hazard forecasting and response planning. The methodology to identify and explore the impact of compound flooding uses national monitoring sources and freely available numerical models allowing application to other estuaries. The results presented demonstrated the application using a flashy catchment, representing the most vulnerable estuary setting in the UK. The results show the spatial variability in inundation area in response to different flood driver combinations, while the discussion provides insights to improve hazard warning and investigate future uncertainty in such catchment types.

The authors use of impact information (e.g., bus cancellations) is an innovative way to develop a historic record of flood events in the absence of documented records. The figures are clearly presented, and all are thoroughly discussed in the text to illustrate key messages/findings. The contour plots Figs 6-8 are a clear way to illustrate compound hazard impact and compare results, which could be applied to different studies to explore multiple hazard drivers.

Thank you for these comments, and we are pleased that the application of the method and novel approaches have been recognised here. The comments help to improve the clarity and readability of the figures, and the queries have provided an opportunity to consider the usability of the results for local authorities and identify future research questions.

Minor queries/comments:

The hazard thresholds use sea level and river discharge. I assume the elevation and discharge choice was to reflect the long-term monitoring available, so the results can be used by authorities in future assessments using available data. A question for consideration in the reply to reviewers is: would there be any value in using two elevation metrics in the analysis?

As the reviewer identifies, sea level (m) and river discharge ($m^3$/s) are used as flood hazard drivers in the analysis and numerical model due to availability of data. River discharge ($m^3$/s) is an appropriate metric for two reasons: (1) as a boundary condition for the model it ensures that the correct volume of freshwater is inputted (should there be any discrepancies between modelled and measured river cross-sectional area); and (2) discharge is a transferable metric to other systems and the outputs are therefore meaningful across catchments.

Elevation would also be a practical metric in the context of the river levels; local authorities could more easily assess and review rising river levels against a stage / river level threshold for flooding, as opposed to a river discharge.

In the abstract it should be clarified the time lag considered is 3hrs. Sensitivity to a range of time-lags is not considered.

Yes, this is a fair point, and the abstract has been edited:

> L22: and sensitivity (~7%) due to a 3-hour time-lag between the drivers.

When mentioning locations in the introduction (e.g., Lancaster and Humber) they should be located as NW or NE England.

Yes, this is a good point, this has been added to the manuscript.

L110, clarify if the record length is determined by the monitoring duration or the start of this study (using available data when the simulations were performed).

The instrumental record length (from the tide gauge at Llandudno and river gauge at Llanwrst) is determined by the monitoring duration, up to the point when the analysis was performed. Instrumental data was only included in the analysis when paired TWL and Q observations were recorded at the gauge at the same time.

The following text has been added to the manuscript:

*L129: The record length used in the analysis here is determined by the monitoring and modelling duration.*

L130, 22:00 is pm the day before the peak river discharge. It's not clear if the flooding occurred on the falling max tide or the incoming tide.

The recorded flood event occurring on 22 November 1980, as described on Line 130, is an interesting event to explore in more detail. The Natural Resource Wales database of Recorded Flood Extents identifies the start date of this event as 22 November 1980, and an end date 23 November 1980. The Qmax is 433 $m^3$/s, and peak river discharge coincides with low tide on 22 November 1980, 05:00. The record of historic flooding describes this event as a fluvial flood, from the main river, caused by overtopping of defences. The location of flooding is described as 'Conwy Valley, Cae'r Groes, Llanrwst'.

The record does not provide enough detail to identify if the flooding happened on the falling tide or rising tide, but this may not be an important consideration in this event as it is considered a fluvial event.

The manuscript has been edited and the following text added to reflect that this is described as a fluvial event, and there is a lack of information to identify if compound flooding occurred as the exact timing of the flooding is not provided.

*L140: Figures 1c and 1e show the 22 November 1980 flood event where Qmax was recorded as 428 $m^3$/s at 03:45 am. TWLmax was 4.5 m at 22:00 am (which included a 0.25 m skew surge) however lack of exact information on the timing of the flooding mean it is difficult to say if TWLmax contributed to flooding, and this is a compound flood. The NRW catalogue notes that there was widespread fluvial flooding in the Conwy Valley at this time, although since this was the pre-internet era there are no further online records.*

Figure 2. Can Caesar-Lisflood provide information about the currents within the estuary to understand when slack tide occurs relative to the river flow? This would help

understand the processes involved in holding the water within the estuary domain. The time lag is currently presented between Qmax and TWLmax. There could be asymmetries that prevent slack water occurring at TWLmax. It would be interesting to see the Qmax lag relative to slack tide (turning from flood to ebb). It would be worth checking the colour bar doesn't constrain the lag so it is relative to the TWLmax tide. The compound flooding could occur on the next tide (more than +6hrs might represent flood tide again). If there are events that occur more than +/- 6hrs either side of TWLmax can they be identified (two panels maybe)?

We did not explicitly simulate the events that are presented in Figure 2. The data presented in Figure 2 is an analysis of the instrumental data available from Llandudno tide gauge and Cwmlanerch river gauge and it is not possible to classify the lag time relative to slack water, as we don't know the time of slack water for these observed events. That is not captured in the instrumental records.

Events with a lag time greater than +/- 6 hours are not included in Figure 2; as the reviewer explains, it would be hard to assign a Qmax to a specific event.

Despite this, the comment is an interesting one and would be an interesting metric to consider when analysing compound events. Therefore we have added a comment to the discussion to reflect that the research could be developed to explore the lag time between Qmax lag relative to slack tide (turning from flood to ebb) using a 3D baroclinic model. Caesar LISFLOOD can provide information about flow velocity in fluvial and estuary settings however it is two dimensional (depth averaged) and may lack the process representation to fully resolve meaning a 3D hydrodynamic model may be better.

*L711: The lag time is currently presented as between Qmax and TWLmax, however there could be asymmetries within the estuary that prevent tidal slack water occurring at TWLmax. The Qmax lag relative to slack tide (e.g. turning from flood to ebb) could be explored, however significant 3D lateral flows in the Conwy Estuary (e.g. Robins et al., 2012; Howlett et al., 2015) would mean that identifying location and timing of slack water would require a 3D baroclinic model.*

Howlett, E.R., Bowers, D.G., Malarkey, J., Jago, C.F. (2015) Stratification in the presence of an axial convergent front: Causes and implications. *Estuarine, Coastal and Shelf Science.* 161, 1–10.

Robins, P.E., Neill, S.P., Giménez, L. (2012) A numerical study of marine larval dispersal in the presence of an axial convergent front. *Estuarine, Coastal and Shelf Science.* 100, 172–185.

Figure 3 and 11. Using a different shape to explain the colour coding in legends (e.g., squares rather than circles) would help quickly separate this information from the circles used to indicate the top 50 Q.

Yes, this is a good suggestion. The legend in Figure 3 and 11 have been updated to include a panel of colour to represent recorded and not recorded flood event, as shown below:

[Figure]

[Figure]

L366, the flood extents aren't clearly assigned to the results in Figure 9. How were the three scenarios in each subplot selected? I assume it was to illustrate a similar FloodArea as defined on L365. In Figure 9 the distance up estuary could be marked (or the place names added) to show the locations of sensitivity in Figure 10. To help show how the icon boxes (Fig 9) link to the results the grid could be plotted on the TWLmax and Qmax axis, with the grid boxes labelled as tide dominant, river dominant, compounded and moderately compounded. Putting the grid onto Figure 11, would help show the probability of occurrence of the different flood drivers.

Thank you for highlighting this. The lateral flood extents in Figure 10 shows 12 model scenarios, with three selected to show sensitivity of FloodArea to small changes in Qmax and TWLmax for TWL, Q, moderate compound, and extreme compound scenarios. The three scenarios in each subplot were selected to be representative of FloodArea across the estuary for each type of scenario. For example, Figure 10a demonstrates that FloodArea is sensitive to Qmax when TWLmax remains the same; higher Qmax creates a greater FloodArea.

Different scenarios could be presented here (e.g. $Q_{3-4}TWL_{12}$), and the sensitivity illustrated in Figure 10 would have been the same. These scenarios were selected because they most clearly demonstrate the sensitivity of FloodArea to the drivers.

The four scenarios presented in Figure 9 are selected to represent FloodArea produced by different combinations of drivers.

The following changes have been made to the manuscript:

*L415: For each case (i-iv), three simulations were presented to illustrate sensitivity of FloodArea for scenarios with similar driver magnitude: (i) TWL dominated, 3.1 - 6.5 $km^2$, (ii) Q dominated, 11.13 - 11.8 $km^2$, (iii) moderate compound, 5.4 - 8.3 $km^2$, and (iv) extreme compound, 8.8 - 9.1 $km^2$.*

Distance along the estuary (N-S) follows the channel, rather than as the crow flies. Therefore individual distance markers have been added to Figure 9a along the channel, rather than a N-S axis next to the figure.

Place names have been added to Figure 9b, that relate to the places named in Figure 10.

Plotting the grid on the TWLmax and Qmax axis, representing the complete 13 x 14 grid, makes the figure look busy. This is why we have kept the simplified 3 x 3 grid on Figure 9. The suggestion to label the grid boxes is a good one. This was stated in the Figure caption, but has now also been added to Figure 9. Please see the updated Figure below.

[Figure]

P20/21, check "FloodArea" is always in italics.

Checked and edited where necessary.

Figure 11, should P=0.9 in the caption be 0.6 to match the contours plotted? For the joint probability it needs to be clarified that co-concurrence means Qmax must occur in +/- 6hrs of TWLmax (for a flashy estuary), they do not have to occur at exactly the same time.

Thank you for highlighting this. Yes, the caption for Figure 11 has been edited to show that P=0.6 is the lowest probability displayed.

L684, have not.

This has been edited.

Author contributions, should CN be NC?

This has been edited.

**Response to reviewer 2**

The manuscript presents an interesting framework for setting thresholds for estuarine compound flooding using combined hydrodynamic statistical techniques. The approach is applied to Conwy Estuary, North Wales, a particularly vulnerable area to compound flooding. It represents a current thematic area and can be particularly useful in improving compound flooding assessment in estuaries worldwide. The study's main objective is to identify the coastal and fluvial conditions that lead to flooding in an estuarine system.

The manuscript is well-written (English level) and presents high-quality scientific content with interesting results and discussions. No doubt, a lot of work has been done, and the research is highly relevant. However, the connection between chapters (structure) needs improvement. In addition, the authors make some general assumptions throughout the text about specific terms that can confuse the reader, especially if he/she is not familiar with the United Kingdom or the field of coastal/estuarine flooding (especially in the Introduction and Methods Section). Further explanations need to be given in these specific points to improve the reader's comprehension for a broad scientific audience.

Therefore, I do not recommend publishing this manuscript before major revisions.

Thank you for these comments, and we are pleased that the reviewer acknowledges the '*high-quality scientific content'*, '*interesting results and discussions',* and the relevance of the research to improve compound flooding assessments in estuaries worldwide. We provide responses to each comment below. The comments help to improve the context and justification for the research, as well as highlight the novelty of the research.

My main critics are the following:

1.  In Section 1 (Introduction), the importance of the topic is well developed, although some key elements need some clarification. For example, the authors talk about thresholds being fundamental to assessing compound flooding; however, they do not explain what a threshold is or why they are important from a technical point of view.

We appreciate that the introduction would benefit from an explicit definition of a flood threshold in the context of this research. Flood thresholds are already introduced in Section 1:

*Flooding can occur when one or several of these processes cause water levels to exceed a critical threshold, such as a sea defence (EA, 2022).*

And:

*Statistical analyses of long-term data, e.g., from paired coastal and riverine gauge observations can show dependence between these drivers (Hendry et al., 2019; Camus et al., 2021; Lyddon et al., 2022) and can be used to examine the joint exceedance probability of estuary water levels based on when marine and terrestrial drivers are above the predefined thresholds (e.g., 95th or 99th percentile) (Kew et al., 2013, Salvadori et al., 2016).*

We appreciate that terms can be understood differently, and it is valuable to define the interpretation of the threshold used in this research. A threshold is used here to define the river and coastal conditions under which flooding could occur, and the conditions at which

a decision or action should be taken to mitigate the flood hazard e.g. to issue warnings. The following text has been added to section 1 to clarify this:

*L46: A threshold represents a meteorological, river and/or coastal condition at which flooding hazard increases (Sene et al., 2008). If a forecasted storm event could exceed the threshold then action to mitigate the hazard should be taken, for example, issue a flood warning.*

*Sene et al. (2008). Thresholds. In: Flood Warning, Forecasting and Emergency Response. Springer, Berlin, Heidelberg. https://doi.org/10.1007/978-3-540-77853-0_3*

2. In section 2 (Methods), much information is given; however, some details and the connection between the subsections must be improved. For instance, why the authors used the recorded data (subsections 2.2 and 2.3) is not well explained. What is the objective of collecting that information? Is it for validation of the hydrodynamic simulations? A clear statement (short paragraph) should be written right at the beginning of the subsection, as it was done for subsections 2.4 and 2.5. Subsection 2.6 needs the same explanation at the beginning of the paragraph.

We appreciate that additional justification for exploiting the recorded data could be provided. This step of the methodology aims to identify if flooding has occurred and been recorded in the Conwy Estuary as a result of the extreme TWL and Q events identified in section 2.2 (displayed as squares and circles in Figure 2). The recorded data includes online newspaper articles, local authority reports, or flood reports, to help identify if flooding happened as a result of extreme coastal and/or river conditions. This methodology creates a more thorough and comprehensive record of historic flooding events, to help to identify a flood threshold.

This step of the methodology is introduced in section 2.2:

*L197: It is important to know whether all of these extreme events in fact caused flooding as one might expect, and which other extreme events in the record led to flooding, to be able to establish meaningful thresholds for flood warning.*

And section 2.3:

*L210: Web scraping approaches (also referred to as web extraction or web harvesting) were used to evaluate whether there is further evidence of recorded flooding in the Conwy estuary within the 100 extreme Qmax and TWLmax events plotted in Figure 2.*

The following text has been added to the start of section 2.3 to reinforce the aim of this step of the methodology:

*L208: Records of historic flood events were expanded by exploring internet records. Online resources were used to identify if flooding happened as a result of extreme coastal and/or river conditions to create a more comprehensive record of historic flood events.*

An introductory sentence has been added to the start of section 2.6 in the manuscript:

*L372: The following methodology was applied to identify the flood extent under each scenario generated in section 2.5.*

3.   The authors use specific terms not defined anywhere in the text—for example, skew-surge, storm surge, total water level, flood event. Although familiar to the coastal flooding research public, they must be defined the first time they are mentioned in the manuscript.

We appreciate that readers may not be familiar with these standard terms, and some expressions can be understood differently. However, we do not believe that standard terms need to be defined in every manuscript. Therefore, the following statement has been added to section 1 of the manuscript to refer to a specific standard that defines coastal and hydrological terms:

*L44: Standard terms follow the definitions outlined in Pugh (1987).*

*Pugh, D.T. (1996) Tides, surges and mean sea-level (reprinted with corrections), Chichester, UK. John Wiley & Sons, Ltd*

*Chow, V.T., Maidment, D.R. and Mays, L.W. (1988) Applied Hydrology. International Edition, McGraw-Hill Book Company, New York.*

We appreciate that clarification of how a **flood event** is interpreted in the context of this research is valuable. This has been clarified in the introduction, as being an occasion when coastal or river conditions lead to flooding, in section 1:

*L49: Estuaries are particularly vulnerable to the effects of compound flood events when coastal and fluvial drivers can occur concurrently or in close succession to generate flooding.*

And in Section 2.2:

*L131: Natural Resource Wales (NRW) has collated information on Recorded Flood Extents to show areas that have flooded in the past from rivers, the sea or surface water, …. The database of polygons (Figure 1a) shows 22 Recorded Flood Extents in the tidally-influenced Conwy estuary. Of these Recorded Flood Extents, 14 events were driven by high sea levels or river flows or both that caused flooding by channel capacity exceedance or overtopping of defences (i.e., ignoring flooding due to obstructions, blockages, local drainage issues, and excess surface water was ignored).*

4.   In sections 2.1 and 2.2, several flooding events are described; however, a general description of storm surge climate and tidal regime in Conwy Estuary is missing.

Additional information on the tidal regime and storm surge climate in the Conwy Estuary has been added to the manuscript:

*L103: The estuary is macrotidal, which is common for the UK, with a 4-6 m tidal range. The semi-diurnal tide displays pronounced tidal asymmetry, characterised by short, fast flood tides and longer, slower ebb tides, which is typical of many macrotidal estuaries. Current speeds reach 1.3 m s⁻¹ during the 2.75 hr flood, while ebb current speeds are 25-30% smaller (Jago et al., 2024). The estuary is subject to the effects of surge generating, low pressure Atlantic storms, elevating sea level up to 1.6 m above predicted levels.*

5.   It is hard to follow section 2.2. and the reasoning behind the selection of flooding events. The authors mentioned that only six flood events were recorded in TWL and river discharge; however, further in the text, they mentioned the top 50 TWL and Qmax events. The authors need to make this explanation more clear. In addition, the authors also need

to explain better what was considered a flood event. This general information should be shown at the beginning of the section, followed by a detailed description of single events already presented in the text.

We have now added text in section 2.2 to clarify that the data presented in Figure 2 shows six Recorded Flood Extents as well as the top 50 extreme river events and top 50 sea level events from the 40-year gauge records. Only some of these extreme events from the gauge records will have caused flooding, some will not, but we wanted to show the Recorded Flood Extents within the context of extreme events that have occurred and highlight that there are potentially many events that have caused flooding that are not recorded.

*L162: Flood drivers Qmax and TWLmax during the six Recorded Flood Extents in NRW's data catalogue are shown as stars in Figure 2. Additionally, from analysis of the ~40 years of river/sea gauge data (see Section 2.1), the top 50 most extreme Qmax and corresponding TWLmax events are shown as circles in Figure 2 (each of these corresponding events occurs within a 'storm-window' of one another, defined as 20.25 hours for the Conwy based on the average duration of event hydrographs over a 30-year period; Lyddon et al., 2021). ... One top 50 Qmax event corresponded with a top 50 TWLmax event, so that 99 extreme events were identified. Not all of these 99 extreme events from the gauge records necessarily caused flooding but this data highlights that there are potentially many events that caused flooding that are not recorded, as explored below. Further, two of the six Recorded Flood Extents corresponded with the 99 extreme events, meaning a total of 103 events are plotted in Figure 2.*

6.  Section 2.4. Is the model solving the equations of movement in 2D or 3D? Please clarify it. In addition, if it is solved in 2D, please justify why it is not solved in 3D and vice-versa.

The following text has been added to section 2.4:

*L243: CAESAR-Lisflood is a geomorphological and landscape evolution model that combines the Lisflood-FP 2D hydrodynamic flow model (Bates et al, 2010) with the CAESAR geomorphic model. Lisflood uses a flow routing algorithm that determines the direction of flow based on the elevation gradient, and conserves mass and partial momentum. CAESAR-Lisflood does not run in 3D, and this functionality is not required to explore flood inundation. Baroclinicity is not an important process to represent for this research, and would require additional computational expense.*

Bates, P. D., Horritt, M. S., & Fewtrell, T. J.: A simple inertial formulation of the shallow water equations for efficient two-dimensional flood Inundation modelling. Journal of Hydrology, 387(1-2), 33-45. https://doi.org/10.1016/j.jhydrol.2010.03.027, 2010.

7.  In subsection 2.5, a general description should be added to the first paragraph of the subsection, explaining the number of scenarios, which are the drivers tested (total water level, storm surge, river discharge, time lag) and the simulation period (72 hours). I know this information is given through the section; however, it is scattered and confuses the reading.

This section summary text has now been included, in Section 2.5:

*L302: Three scenarios, each consisting of 520 simulations, tested the influence of the relative drivers of estuary flooding (tidal water level, storm surge, river discharge, and time lag) – see Table 2 and Figure 5.*

8.  Section 2.8 does not clearly explain why joint probabilities were used. Why is it important? This is partially explained in the introduction and should be remembered here. Similarly, a further explanation of why Copulas is applied from a technical point of view should be given. For instance, are Copulas used to generate synthetic samples of extreme sea levels and river discharges, thus making their respective probability distribution more robust to apply joint probability methods? A general explanation (short and from a practical point of view) should be given about how statistical marginal distributions, copulas, and Bayesian methods are linked to each other.

The reviewer is correct in summarising how copulas are used in this research. The following text has been added to section 2.8 to clarify the inclusion of this methodology:

*L419: Joint probabilities are important in statistics, providing a way to model and analyze the simultaneous occurrence of events. In the context of flood analysis, the joint probabilities identify the likelihood of combinations of coastal and river conditions occurring, and capture relationships between variables (Wu et al., 2021; Olbert et al., 2023; Moradian et al., 2023). The joint probability of river and sea level conditions can be interpreted in the context of i) hydrodynamic model outputs to identify the likelihood of combinations of conditions occurring to create a flood hazard, and ii) recorded historic flood events to provide context to the severity of flood events. Copulas are effective at modelling nonlinear dependence structures and joint distribution between two variables. The copula functions (Sklar, 1959) are used here to generate synthetic bivariate pairs of extreme sea levels and river discharges, thus making their respective probability distribution more robust to apply joint probability methods.*

*Wu, W., Westra, S., Leonard, M.: Estimating the probability of compound floods in estuarine regions. Hydrology and Earth System Sciences. 25(5), 2821–2841, 2021.*

9.  In Section 2.8, there are references to several tables from another paper. I do not believe this is a good way to mention it. If applied, this content should be added to supplementary data or referred to the work without the table number.

We appreciate that it may be unconventional to refer to specific tables from another paper. However, the authors feel it is important to keep the table numbers within the text in section 2.8 to provide valuable detail for the reader to understand the copula-based methodologies. Removing specific references to the table would mask the specifics required to communicate the exact methodology used here, and the authors do not feel it is necessary to reproduce the tables in their entirety within the supplementary information for this manuscript.

10. Section 4 (Discussion) lacks a clear first paragraph, generally stating the manuscript's main findings. The first paragraph is too vague and similar to the conclusion section's first paragraph. I suggest making a general statement on the main findings and then detailing each in the following subsections.

We agree with this suggestion and have now rewritten the first paragraph of section 4 so that the main findings are clearer:

*L631: This research has established site-specific driver-thresholds for flooding in an estuary environment via hydrodynamic modelling. The simulations have been verified and contextualised using documented records of flooding, together with data analysis and statistical analysis of instrumental gauge time series. With application to the Conwy estuary, North -Wales, the inundation model was applied to a series of idealised combined river and sea level compound events. We show that flooding is co-dependent on TWLmax, Qmax, and their relative time lag; and that historic records of flooding can be used to set driver and flood extent thresholds that isolate minor and severe flooding. Below we discuss the thresholds for flooding and the importance of accurate records of historic flooding events. We consider how these thresholds may change under different driver behaviours and combinations, and future climate conditions.*

11.   In Section 4.1, I do not understand why the documented flooding records are discussed if they are not even in the section Results (they are shown in Methods). I suggest shortening it to conclusions or moving it to the end of subsection 4.2, "Thresholds for flooding", where you could link the importance of historical records of flooding events to validate numerical/statistical techniques and calculate optimal thresholds (for instance).

The discussion on 'Documented records of flooding' has been moved to the end of section 4.2, and the following text edited in the manuscript as suggested:

*L731: Documenting compound flood events aids in understanding and analysing the drivers, interactions, and impacts of the hazards (Haigh et al., 2015; Haigh et al., 2017), validating numerical and statistical techniques, and calculating optimal thresholds.*

12.  Section 4 shows several interesting discussion points; however, the authors did not discuss much about the novelty of the method. How innovative is the approach applied here? For instance, Section 4.2 shows a good discussion about setting thresholds from an end-user point of view (flooding mitigation planning); however, the authors do not discuss the hydrodynamic-statistical approach itself.

The manuscript identifies the novel methodology to establish site-specific driver-thresholds for flooding, which applies hydrodynamic and statistical approaches to build process-understanding, and combines model outputs with historic flood records to build knowledge of flood thresholds and impacts. As the reviewer identifies, there is novelty in interpreting statistical and hydrodynamic results in the context of historical records, to validate outputs to support communication for application to flood mitigation planning.

There is novelty in where these approaches are applied (some for the first time). The combined methodology is applied to a flashy catchment, typical for the west coast of the UK and representing the most vulnerable estuary setting in the UK. It is an example of an application of a freely available numerical model to enable application to other estuaries. Multivariate approaches are superior to the traditional univariate assessment methods, and methodology provides a cost-effective, practical approach for delineating compound flood hazards. Due to the computational cost of multivariate statistical modelling approaches, these methodologies are best applied on a case-by-case basis. Therefore, this research represents the application of a robust method for the assessment of coastal-fluvial flooding in a new location.

There is novelty in the model setup as Caesar-Lisflood uses an improved DEM, with recently developed novel techniques in model validation. The Caesar-Lisflood 2D hydrodynamic flow model was combined with a range of publicly available datasets to

represent channel bathymetry, land elevation, location and heights of flood defences and the hydraulic roughness across the model domain (Vasilopoulos et al., 2023). The novelty of the validation techniques is not the focus of this research, and would make the paper unnecessarily long. This is currently being written up into a separate manuscript.

There is novelty in the results, as site-specific driver-thresholds for estuarine flooding have not been identified before. The paper identifies spatial variability in threshold and vulnerability to compound effects within an estuary, and a range forecast for river discharge and sea levels that forecasts need to be particularly accurate.

We believe the novelty of the research is captured well in Section 4, at the start of the Discussion but the following text has also been added:

*L427: The Copula method was employed in this study to compute joint probabilities for extreme sea levels and river flows co-occurring in the Conwy for the first time.*

*L613: Figure 11 represents a novel approach to interpreting joint probabilities in the context of historic storm events, to better understand the relationship between drivers and impacts of flooding.*

*L642: for the Conwy estuary we show for the first time that flooding is co-dependent on TWLmax, Qmax, and their relative time lag.*

*L801: This research developed a novel framework that utilised a combination of historic estuary flooding records, instrumental monitoring data, numerical modelling, and probabilistic analyses to identify driver-thresholds for compound flooding, for an estuary that is especially vulnerable to compound flooding events (Conwy, N-Wales, UK).*

Vasilopoulos, G., Coulthard, T., Robins, P., Lyddon, C., Barkwith, A., Chien, N., and Lewis, M.: Development and validation of flood inundation models for estuaries, EGU General Assembly 2023, Vienna, Austria, 23–28 Apr 2023, EGU23-5858, https://doi.org/10.5194/egusphere-egu23-5858, 2023.

13. To my understanding, the approach presented here relies on having sufficient data (recorded flooding, water level and river discharge observations, good quality topo-bathymetry data), which is not the reality of several regions globally (perhaps also in the UK). I see the authors discussed that in section 4.2; however, the discussion is focused on the UK. I suggest adding a short discussion about the quality of the forecast and data availability on regional (e.g., Europe) and global scales.

This is a good point, and the following text has been added to the discussion:

*L750: The combined approach to identify driver-thresholds for compound flooding presented here, and additional parameters suggested to develop the approach, rely on availability and access to sufficient instrumental data at the appropriate temporal resolution, and topographical and bathymetric data at appropriate spatial resolution. The UK sea levels, river discharges, and topography are recorded, archived, and accessed via national government and research agencies (e.g. British Oceanographic Data Centre, National River Flow Archive, Centre for Environment, Fisheries and Aquaculture Science, and Channel Coastal Observatory). However, nearly 50% of the world's coastal waters remain unsurveyed (IHO C-55, 2021), and 290 tide gauges that form the Global Sea Level Observing System (GLOSS, Holgate et al., 2013) are unevenly distributed across the globe and do not account for local, vertical land movements. The approach described here could supplement existing observation systems with new technologies to improve records of coastal processes (Marcos*

et al., 2019), at local scales including X-band radar-derived intertidal bathymetries (Bird et al., 2017), X-band radar-derived tide and surge (Costa et al., 2022), and regional scales including Satellite-Derived Bathymetry (Cesbron et al., 2021 and Hasan and Matin, 2022), and satellite altimetry (Cipollini et al., 2019), which measures the sea level from space with sufficiently dense global coverage. Global model projections of storm surge and tide can be downscaled and applied to inform assessment of coastal flood impacts (Muis et al., 2023). Temporal and spatial gaps also occur in the global river discharge observing network, and hydrometric data are not available in real time (Lavers et al., 2019; Harrigan et al., 2020). Research has focused on coupling surface and sub-surface runoff models, hydrologic models, and land surface models, forced with global atmospheric reanalysis (e.g. ECMWF's ERA5) to produce river discharge reanalysis (Harrigan et al., 2020). Combining observation and downscaled modelled data to explore thresholds for estuarine flooding is one approach to applying this methodology worldwide.

Bird, C.O., Bell, P.S., Plater, A.J.: Application of marine radar to monitoring seasonal and event-based changes in intertidal morphology. Geomorphology. 285, 1–15, https://doi.org/10.1016/j.geomorph.2017.02.002, 2017.

Cesbron, G., Melet, A., Almar, R., Lifermann, A., Tullot, D., Crosnier, L.: Pan-European Satellite-Derived Coastal Bathymetry—Review, User Needs and Future Services. Frontiers in Marine Science. 8, https://doi.org/10.3389/fmars.2021.740830, 2021.

Costa, W.L.L., Bryan, K.R., Coco, G.: Modelling extreme water levels using intertidal topography and bathymetry derived from multispectral satellite images. Natural Hazards and Earth System Sciences. 23(9), 3125–3146, https://doi.org/10.5194/nhess-23-3125-2023, 2023.

Cipollini, P., Calafat, F.M., Jevrejeva, S., Melet, A., Prandi, P.: Monitoring Sea Level in the Coastal Zone with Satellite Altimetry and Tide Gauges. Surveys in Geophysics. 38(1), 33–57, https://doi.org/10.1007/s10712-016-9392-0, 2016.

Harrigan, S., Zsoter, E., Alfieri, L., Prudhomme, C., Salamon, P., Wetterhall, F., Barnard, C., Cloke, H., Pappenberger, F.: GloFAS-ERA5 operational global river discharge reanalysis 1979–present. Earth System Science Data. 12(3), 2043–2060, https://doi.org/10.5194/essd-12-2043-2020, 2020.

Hasan, G.M.J., Matin, N.: Intertidal bathymetry and foreshore slopes derived from satellite images for static coasts. Regional Studies in Marine Science. 51, 102233, https://doi.org/10.1016/j.rsma.2022.102233, 2022.

Holgate, S.J., Matthews, A., Woodworth, P.L., Rickards, L.J., Tamisiea, M.E., Bradshaw, E., Foden, P.R., Gordon, K.M., Jevrejeva, S., Pugh, J.: New Data Systems and Products at the Permanent Service for Mean Sea Level. Journal of Coastal Research. 29(3), 493, https://doi.org/10.2112/JCOASTRES-D-12-00175.1, 2013.

IHO C-55: Publication C-55 "Status of Hydrographic Surveying and Charting Worldwide.". Monte Carlo: IHO, https://iho.int/en/iho-c-55, 2021.

Lavers, D., Harrigan, S., Andersson, E., Richardson, D. S., Prudhomme, C., and Pappenberger, F.: A vision for improving global flood forecasting, Environ. Res. Lett., 14, 121002, https://doi.org/10.1088/1748-9326/ab52b2, 2019.

Marcos, M., Wöppelmann, G., Matthews, A., Ponte, R.M., Birol, F., Ardhuin, F., Coco, G., Santamaría-Gómez, A., Ballu, V., Testut, L., Chambers, D., Stopa, J.E.: Coastal Sea Level and Related Fields from Existing Observing Systems. Surveys in Geophysics. 40(6), 1293–1317, https://doi.org/10.1007/s10712-019-09513-3, 2019.

*Muis, S., Aerts, J.C.J.H., Á. Antolínez, J.A., Dullaart, J.C., Duong, T.M., Erikson, L., Haarsma, R.J., Apecechea, M.I., Mengel, M., Le Bars, D., O'Neill, A., Ranasinghe, R., Roberts, M.J., Verlaan, M., Ward, P.J., Yan, K. (2023) Global Projections of Storm Surges Using High-Resolution CMIP6 Climate Models. Earth's Future. 11(9), https://doi.org/10.1029/2023EF003479, 2023.*

14.  In section 5 (Conclusions), the authors wrote a good introductory first paragraph; however, just after, they talk about historical floods, which is not the paper's main goal (I suggest deleting this paragraph or shortening and moving it to the end of the section). I would have expected that they answered the main objective straight after the introductory paragraph (to identify the coastal and fluvial conditions that lead to flooding in an estuarine system).

The concluding comments about the historical flood records are still valuable and important outcomes of the research to create a more comprehensive flood record, so we have followed the reviewer's suggestion to move this paragraph towards the end of the conclusion.

SPECIFIC COMMENTS

15.  Lines 6 – 11. Please replace "UK" with "United Kingdom".

Thank you, but the authors follow the journal's convention where the abbreviations (i.e. UK and USA) are frequently used.

16.  Line 14. Please replace "UK" with "United Kingdom".

See Comment nº 15.

17.  Line 19. Please replace "N-Wales" with "North Wales".

Thank you, the author amended the text following your suggestion.

18.  Lines 20–22. It is not clear what was amplified. What does sensitivity 7% mean?

Lines 20-22 refer to flood extent throughout. The flooding extent is amplified with compounding coastal and river conditions, and flood extent increases by up to 7% when the time lag between peak river discharge and the total water level is adjusted by -3 hours. The authors have added one additional reference to flood extent in this sentence for clarification.

19.  Line 30. Replace "–" (en-dash) with "—" (em-dash).

Thank you, we have replaced the en-dash with the em-dash in this instance. We will allow the journal typesetter to handle the remaining occurrences of this grammatical preference in line with journal policy.

20.  Line 34. I understand the UK refers to the United Kingdom; however, the authors should define all the acronyms. This acronym has not been previously defined in the text. Suggestion: "United Kingdom (UK)". Then, after that, you could only use the UK.

See Comment nº 15.

21.  Line 80. "Modelling statistical and probabilistic methods," wouldn't they be the same?

These are two separate methods. Statistical modelling focuses on capturing relationships between variables and fitting a model to these, while probabilistic modelling places emphasis on understanding uncertainty using distributions.

22.  Line 89. "N-Wales". Please homogenise the use of "North Wales". If "N-Wales" is used, then it needs to be defined, for instance: "North Wales (N-Wales)….". Then, use only N-Wales after that.

The text has been amended following your suggestion. This also solved Comment nº 17.

23.  Line 110. In "November 1980 - February 2023", a dash is inappropriate. En-dash "– " should be used for ranges of dates and numbers. Please replace throughout the document whenever applicable.

The dash has been resolved on line 110, and see Comment nº 15.

24.  Lines 126–133. The terms "total water level (TWLmax)", "predicted tide level", "skew surge", and "storm surge" are not defined. For instance, does the "total water level" include the sea-level anomaly/trend? If so, the sea level trend could interfere with your results. This should be well clarified.

See comment nº 3 in respect to definitions of standard terms.

The question about sea level trends is a good one. The total water level time series from the Llandudno tide gauge was linearly detrended to remove the historic sea level trend. The following text has been added to section 2 of the manuscript to reflect this:

*L122: Total water level from the Llandudno tide gauge was linearly detrended to remove the effects of a historical sea level trend from the time series (Coles 2001).*

25.  Line 130. Please define the acronym "NRW" and homogenise the use of Natural Resource Wales. Use "Natural Resource Wales" or NRW throughout the document (e.g., line 138).

Thank you, the author amended the text following your suggestion.

26.  Figure 1. The labels of the x-axis in panels (e) and (f) do not follow the same pattern. Please homogenise them.

This x-axis in Figures 1e and 1f has been edited for consistency.

27. Line 150. What is "event hydrographs"? Please define it.

This has now been described as referring to extreme hydrograph events in Section 2.2:

*L163: Additionally, from analysis of the ~40 years of river/sea gauge data (see Section 2.1), the top 50 most extreme Qmax and corresponding TWLmax events are shown as circles in Figure 2 (each of these corresponding events occurs within a 'storm-window' of one another, defined as 20.25 hours for the Conwy based on the average duration of extreme event hydrographs over a 30-year period; Lyddon et al., 2021).*

28. Line 177. Please change "however," to "; however, …" or ". However, …"

This has now been changed.

29. Line 188. I am not used to the term "Web scraping approaches". Is that a proper term to be used in a scientific paper? Maybe change it to "Web searching"?

Web scraping is the common term in peer-reviewed literature, but it is also known as web extraction or harvesting and the text has been modified to show this.

30. Table 1. It seems to be cut at the bottom of the preprint. Please check if this is indeed the case. Bellow a screenshot:

| Label | Code |
|-------|------|
| 0 | None |
| 1 | River discharge |
| 2 | Storm surge |
| 3 | High tide |
| 4 | Storminess |

This is a formatting issue in the pre-print, and will be addressed in the revised manuscript. The original table is formatted to include a border around all four edges.

31. Line 201. In the sentence "… with yellow dots indicating there is evidence of flooding and blue dots indicating there is no evidence of flooding." How can it not have evidence of flooding and be on the internet? I couldn't understand it.

This sentence has now been rephrased:

*L224: The web searches isolated an additional 26 recorded floods that matched extreme events in our analysis, as shown in Figure 3, with yellow circles indicating these 26 events. The blue circles in Figure 3 indicate extreme events where there was no online evidence of flooding.*

32. Lines 209–210. ".... leave uncertainty in where to set driver thresholds and patterns for flooding, especially for less extreme Qmax and TWLmax that led to compound flooding.". The concept of threshold (quantiles, peaks-over-threshold, block maxima) and event definition (How long an event was considered to last? Was used any declustering schemes?) needs to be clearly described in the introduction or previously in the methods section.

The Qmax and TWLmax variables are now clearly defined in Section 2.2:

*L137: The behaviour of the drivers of the six Recorded Flood Extents was reconstructed from the sea level and river flow data records, including timing and magnitude of peak river discharge (Qmax), total water level (TWLmax), predicted tide level, and skew surge that preceded the flood.*

The definitions of flooding are also clearly described in section 2.2:

*L133: Of these Recorded Flood Extents, 14 events were driven by high sea levels or river flows or both that caused flooding by channel capacity exceedance or overtopping of defences (i.e., ignoring flooding due to obstructions, blockages, local drainage issues, and excess surface water).*

…and in section 2.3:.

*L208: Online resources were used to identify if flooding happened as a result of extreme coastal and / or river conditions to create a more comprehensive record of historic flood events.*

…and finally the simulated *FloodArea* was defined in section 2.6 (equation 1).

33. Line 213. I do not fully understand what the authors considered a top 50 Qmax and TWLmax. The authors mentioned that only a few recorded flooding were identified (6 of NWR and 20+ in web search). However, in Figures 2 and 3, the authors show more events than that. Top 50 events mean that you have selected the top 50 events, or are you taking the events above the 50% percentile? Please make it more clear.

We have now rewritten section 2.2 so that the definitions are clearer:

*L137: The behaviour of the drivers of the six Recorded Flood Extents was reconstructed from the sea level and river flow data records, including timing and magnitude of peak river discharge (Qmax), total water level (TWLmax), predicted tide level, and skew surge that preceded the flood.*

*L162: Flood drivers Qmax and TWLmax during the six Recorded Flood Extents in NRW's data catalogue are shown as stars in Figure 2. Additionally, from analysis of the ~40 years of river/sea gauge data (see Section 2.1), the top 50 most extreme Qmax and corresponding TWLmax events are shown as circles in Figure 2 (each of these corresponding events*

*occurs within a 'storm-window' of one another, defined as 20.25 hours for the Conwy based on the average duration of event hydrographs over a 30-year period; Lyddon et al., 2021). ... One top 50 Qmax event corresponded with a top 50 TWLmax event, so that 99 extreme events were identified. Not all of these 99 extreme events from the gauge records necessarily caused flooding but this data highlights that there are potentially many events that caused flooding that are not recorded, as explored below. Further, two of the six Recorded Flood Extents corresponded with the 99 extreme events, meaning a total of 103 events are plotted in Figure 2.*

34.  Line 224–226. The main sources for DTM, bathymetry, and flood defence locations should be mentioned in the main text.

The data sources for DTM, bathymetry, and flood defence location are now provided in the text in section 2.4.1.

*L253: The domain topography was based on the marine DEM, Lidar DTM and OS Terrain 5m DTM, all available through Digimap ([https://digimap.edina.ac.uk/](https://digimap.edina.ac.uk/)).. The Lidar DTM data was used to check and, where necessary, augment the flood defences vector database, obtained from the NRW data catalogue ([https://datamap.gov.wales/](https://datamap.gov.wales/)). The processing steps undertaken to produce the model domain are described in Supplementary Information S1.*

35.   Line 245–248. How exactly did the authors gradually adjust the channel bed elevations? Manually editing the bathymetry? The Neal et al. (2022) work should be better described. One or two short sentences should be enough. Also, what is a "stepwise manner"?

The approaches used to adjust the channel bed elevations have been clarified in the manuscript in section 2.4.2:

*L276: We approximated the correct channel bathymetry by manually adjusting the channel bed elevations, re-running the simulation and comparing simulated and observed water levels. We repeated this process until we reached a satisfactory agreement between observed water levels and model predictions at the three gauges. With this method the bed profile is adjusted until it simulates the observed water profile, taking into account flow non-uniformity (Neal et al., 2022).*

36.  Line 249. Why did the authors use two scores (RMSE and Kling-Gupta Efficiency)? Is there any advantage to using that? Formulas should be added to the supplementary material.

Only RMSE is used in the manuscript to test model accuracy, and any mention of KGE in the manuscript and supplementary information has been removed. The equation for RMSE has been added to supplementary information.

*The following equation was used to calculate RMSE scores for flood peaks, which is a widely used metric for evaluating model performance:*

$$RMSE = \sqrt{\frac{1}{n}\sum_{i=1}^{n}(y_i - \hat{y}_i)^2}, \quad (1)$$

*where N is the number of data points, y(i) is the i-th measurement, and y ̂(i) is its corresponding prediction.*

37.  Line 253. "in the upper estuary". Please add the names of the stations in parentheses.

Pont Fawr has been added in parenthesis to clarify the station name.

38.  Line 254: "tributaries". Please explain.

This section of the manuscript has been edited to clarify what is meant by tributaries in section 2.4.2:

*L289: Higher RMSE values in the upper estuary (Pont Fawr gauge) could be attributed to the omission of tributaries in the model that flow into the Conwy downstream of the Cwmlanerch gauge (upstream boundary of the model). These inputs are, as a result, not represented in the discharge data forcing the model. Nevertheless, the set-up remains suitable for the purposes of this research.*

39.  Line 270. "The M2 tidal constituent has an amplitude of 2.71 m and was used to produce a constant sinusoidal curve for 72 hours". Why only the M2? Are the shallow-water harmonics not important in this estuary? Why 72 hours?

This section has now been rewritten to address these questions and make the descriptions clearer, in section 2.5:

*L300: The idealised model scenarios were used to add more detail to the historic records of flooding and instrumental data (Figures 2 and 3) to enable driver thresholds for flooding to be established. Three scenarios, each consisting of 520 simulations, tested the influence of the relative drivers of estuary flooding (tidal water level, storm surge, river discharge, and time lag) – see Table 2 and Figure 5. The simulations consisted of 40 river discharge conditions with incrementally increasing Qmax, in combination with: (Scenario-1) 13 incrementally increasing tide levels combined with a maximum storm surge; (Scenario-2) 13 incrementally increasing tide levels combined with a mean storm surge; and (Scenario-3) 13 incrementally increasing tide levels combined with a maximum storm surge and a three-hour time lag. In total, 40 (Qmax) × 13 (TWLmax) × 3 (scenarios) = 1,560 discrete simulations were performed. Each simulation was run for a period of 72 hours, allowing for model spin-up (thus allowing the assumed initial condition to become consistent with the hydrodynamic system) and with TWLmax and Qmax occurring after ~40 hours. These boundary conditions are described in more detail below.*

*L325: The boundary conditions for total water level consisted of 13 time series for each of the three scenarios. These time series were created using idealised tidal signals combined with residual surges. Firstly, a sinusoidal elevation with a period of 12.42 hours (equivalent to the dominant M2 tidal constituent) was created. This was parameterised to represent mean neap tides at Llandudno. Mean spring and neap tidal amplitudes and high tide levels were determined using a harmonic analysis (T-Tide; Pawlowicz et al., 2002) based on 12 months*

*of tide gauge data from Llandudno (2002-2003). A subsequent tidal prediction revealed that mean high water neap tides reach 1.82 m (OD) and mean high water spring tides reach 3.6 m (OD). The elevation time series was then reproduced 13 times, each time by successively increasing the amplitude so that high water was incrementally increased by 25 cm until equivalent to spring high tides. This experimental design purposely neglected the influence of other constituents so that the results were standardised. The model simulated the shallow water propagation of the tide advancing up the estuary.*

40.  Line 272. "scale factor of 25 cm…. thus creating 13 water level time series". Sorry, I could not understand the reasoning. Between 1.82 m (high neap tide) and 3.6m (high spring tide) and a scale factor of 25 cm (adding 25 cm to 25 cm), I could only count seven water level time series and not 13. Please clarify it.

This section has now been rewritten to address these questions and make the descriptions clearer, see the response above (#39).

41.  Line 276. What is a representative surge shape?

This has now been clarified in section 2.5.2:

*L340: The shape of the surge was representative of typical storm conditions for Llandudno (Environment Agency, 2016), as shown in Figure 5.*

42.  Line 297. What does "spin-up" time mean?

The spin-up time refers to the time taken by the model to allow the assumed initial condition to become consistent with the hydrodynamic system. We have added this brief explanation to the manuscript in section 2.5:

*L307: Each simulation was run for a period of 72 hours, allowing for model spin-up (thus allowing the assumed initial condition to become consistent with the hydrodynamic system) and with TWLmax and Qmax occurring after ~40 hours. These boundary conditions are described in more detail below.*

43.  Line 306. Where exactly does the 40 discharge time series come from? Please clarify.

This section has now been rewritten to clarify the methodology to generate 40 idealised discharge time series. This is included in section 2.5.1:

*L312: The following method was undertaken to generate 40 idealised discharge time series parameterised on the hydrology of the Conwy. Firstly, a two-parameter gamma distribution was used to generate a synthetic series of normalised, idealised gamma curves, that represent hydrograph shapes that cover the natural range of river flow behaviours experienced in the Conwy based on 30 years of river discharge data from the Cwmlanerch river gauge (see Robins et al., 2018).  The gamma curve with the gradient of the rising hydrograph limb that most closely resembled the average gradient of the top 50 Qmax events analysed in this study was selected. The selected idealised hydrograph had the largest gradient representing the flashiest flow behaviour. The magnitude of the idealised hydrograph was then scaled to a peak discharge Qmax of 25 m³/s (i.e., a relatively small*

*river flow event that will not likely cause flooding), with a base flow of 20 m³/s which represents mean flow conditions. The scaling of Qmax was successively increased from 25 m³/s, in 25 m³/s increments, up to a Qmax of 1000 m³/s (i.e., slightly greater than the maximum recorded event of 901 m³/s), always keeping a base flow of 20 m³/s). This created a realistic range of 40 river discharge event time series that were applied to all three scenarios. For each simulation, Qmax occurred at 40 hours (Figure 5).*

44. Figure 5. Is the y-axis label of the panel (c) correct ("Number of events")? Shouldn't It be "Total water level OD (m)"?

Thank you for pointing this out, we have corrected it accordingly.

45. Line 343. Please replace "(ROI, see Figure 1a),"for "(ROI), see Figure 1a, .."

Thank you, we have replaced this occurrence.

46. Lines 348–358. Please consider joining this paragraph with the previous one.

Thank you, we have joined the two paragraphs.

47. Line 357. Please clarify why 520-simulation parameter space Scenarios 1–3. I understood there were 1560 simulations.

Thank you, 520 simulations for each of the Scenarios 1–3. We have fixed this.

48. Lines 360–366. Please refer to what section will show and discuss these results. It is confusing to the reader to know if you are talking about the simulation scenarios previously described or the spatial analysis of the flooding area. In addition, what does lateral flood extent mean? Please clarify and define it.

We define the lateral extent of flooding as the width of the inundated area in the direction perpendicular to the river channel. This is added to the manuscript in section 3.4:

*L589: The lateral extents of flooding, defined as the width of the inundated area in the direction perpendicular to the river channel, for Scenario-3 for cases (a-d) are presented in Figure 10.*

49. Line 368. Why do the authors want to use joint probabilities? At which part of the method the authors are applying this? Is there anything to do with hydrodynamic modelling? Please make it more clear.

See Comment nº 8.

50. Line 371. "to the data", which data?

The text has been edited to clarify the method was applied to sea level and river flow data.

51.  Line 378. "Table 6 of Moradian et al. (2023)". Please see comment 9. Apply the comment throughout the text.

See response to comment nº 9.

52.  Line 380. Why so many metrics? Please justify.

Each statistical evaluation metric is designed to capture specific aspects of functional performance, and relying on a single metric may not provide a comprehensive understanding of how well a function is performing. Using different metrics for assessment purposes is important; because different metrics provide better insights into the performance of the copula functions. So, for a comprehensive contextual assessment and better comparisons of the copula models and realizing the trade-offs, we need to use a wide range of metrics for the evaluation process. This enhances the reliability of the assessment.

53.  Line 403. "dependence measures". I suggest changing this term to "dependence metrics" or "correlation coefficients" and then removing "Correlation Coefficient" after the name of each coefficient you mentioned just after. Also, the authors need to explain further why dependence metrics are important. What is it used for?

These terms have been renamed "correlation coefficients" in the manuscript in section 2.8.1, line 462.

The correlation coefficients are used to quantify dependencies between river flows and sea levels and are used in this context to investigate the dependence between the two input variables and evaluate the accuracy of marginal distributions.

54.  Line 407. The authors need to explain better why they want to use Bayesian methods and how they did it.

The text in section 2.8.3 refers to a Bayesian method that is usually for model inference and uncertainty quantification. This theory is a common approach for analysing compound events. This uncertainty analysis is not presented here, so the authors agree it is confusing to include the detail in section 2.8.3. Therefore, to avoid confusion, the authors have removed mention of the Bayesian method. The references to research which employs statistical methods to calculate joint probabilities have been added to section 2.8.2 to refer readers to these papers for more information about the applied methodology.

55.  Line 415. "is the probability of A being true and". The subject of the sentence is missing.

See comment n◦ 54. This section has now been removed.

56.  Lines 423–432. Please join these two paragraphs. The second one gets confusing when not directly linked to Figure 6.

Thank you, we have joined these two paragraphs in the revised text.

57.   Lines 446–448. Couldn't it be a question of scale in the graphics? Is the figure showing a normalised plot? If not, you compare different units (m) and (m³/s). I would plot a normalised plot to double-check this question. Please correct it if applicable.

These remarks on "near horizontal" and "near vertical" are made separately, and do *not* imply any comparison between the magnitude of influence of water level and river discharge on *FloodArea*. Thus, the remarks are acceptable here. On the other hand, normalisation to compare different units is a good technique, but as we set off a ranking scheme (1-13 for water level and 1-40 for river discharge), a further normalisation would probably make the reader more confused in switching between these rankings and the normalised value).

58.  Figures 6, 7, and 8. What is OD in the y-axis label? Is it the vertical datum? It should be explained in the figure caption.

The OD Ordnance Datum was mentioned in the caption of Figure 4 and hence was used consistently in Figures 5, 6, 7, and 8. Therefore, the readers can understand the meaning of OD in the context.

59.  Line 491. Sometimes, the authors use FloodArea in italics, and sometimes they do not use it. Please homogenise it throughout the manuscript.

Thank you, we revised it so that all *FloodArea*s are in italics.

60.  Line 519. Please replace "9d" with "Figure 9d".

Thank you, we have corrected this.

61.  Line 524. Please explain the terms $Q_1TWL_{13}$, $Q_{40}TWL_1$, $Q_{20}TWL_7$ and $Q_{40}TWL_{13}$. I could not understand why the authors used them.

These are the cases to illustrate the relative "strength" between the two main drivers: river discharge (Q) and total water level (TWL). Here the numbers (e.g. 1 in $Q_1$ and 13 in $TWL_{13}$) denote the rank of each driver. $Q_1$ is the lowest rank (smallest discharge class) and $TWL_{13}$ is the highest rank (highest water level class).

62.  Line 526. What is "TWLmax: Qmax parameter space"? Is that figure 6? Please refer to the figure or provide further explanation.

Yes, the TWLmax: Qmax parameter space is shown in the background of Figures 6, 7, and 8. We made a clear drawing of this parameter space, together with explaining the "icons" (Comment nº 64).

63.  Line 530. What does "lateral extent of flooding" mean?

*Please see comment n◦ 48.*

64.  Figure 10. Same as Figure 9. A better explanation of the numbers following Q and TWL is needed. Also, the use of the icon is not clear.

*The meaning of these icons is clarified in Supplementary Material.*

*The icons used in Figures 9 and 10 relate to TWL-dominant, extreme compound, moderate compound, and river-dominant scenarios in the parameter space, identified in Figure S3. The icons indicate the scenarios referred to within the context of the parameter space.*

65.  Lines 569–575. Please see comment n◦ 10.

[comment n◦ 10: Section 4 (Discussion) lacks a clear first paragraph, generally stating the manuscript's main findings. The first paragraph is too vague and similar to the conclusion section's first paragraph. I suggest making a general statement on the main findings and then detailing each in the following subsections]

*This has been addressed in response to comment n◦ 10.*

66.  Line 578. "piecemeal fashion". Is that a scientific term? Please replace it.

*Replaced with "unsystematic".*

67.     Lines 609–611. You can also mention that earth observation records can supplement estuarine topo-bathymetry and geometry data for multiple purposes, including hydrodynamic modelling. Reference suggestions:

*Valentin Heimhuber, Kilian Vos, Wanru Fu, William Glamore, InletTracker: An open-source Python toolkit for historic and near real-time monitoring of coastal inlets from Landsat and Sentinel-2, Geomorphology, Volume 389, 2021. https://doi.org/10.1016/j.geomorph.2021.107830.*

And

*Costa, W. L. L., Bryan, K. R., and Coco, G.: Modelling extreme water levels using intertidal topography and bathymetry derived from multispectral satellite images, Nat. Hazards Earth Syst. Sci., 23, 3125–3146, https://doi.org/10.5194/nhess-23-3125-2023, 2023.*

*These references have been added.*

68.  Section 4.2 is confusing. It seems the authors are introducing new results instead of discussing the current results. I could understand the relevance of the discussion; however, I suggest the authors re-write parts of the section to clarify that new results are not being shown.

Section 4.2 explains that the results presented in section 3 identify driver-thresholds across the whole estuary, however, co-dependent driver-thresholds for flooding will vary at different locations within the estuary. Section 4.2 develops this point of discussion to further utilise hydrodynamic model results and highlight how these can be presented to identify site-specific thresholds. The authors believe Figures 12 and 13 sit well within the context of the discussion, which considers the importance of setting thresholds based on individual flood model cells to most accurately capture co-dependent flood driver dynamics.

69.    Line 616. What is "web scraped tag(s)"? Was it explained anywhere in the manuscript?

Yes, web scraping is mentioned in section 2.3 "Web scraping approaches (also referred to as web extraction or web harvesting) were used to..".

The tags that are referred to on line 616 have been changed to "web-scraped keywords (tags)".

70.   Line 619. I do not follow the statement, "The coastal events (Figure 12c) occur across a range of river discharge combinations, and thresholds may not need to consider this driver". Figures 12 a and b show that flooding events (time lag and river) occur in a similar range of river discharge to coastal events. Please make it more clear.

The text has been edited to clarify this idea:

*L648: The coastal events (Figure 12c) occur under high sea levels and across a range of river discharge combinations, indicating thresholds for flooding in the coastal zone should consider sea level as the dominant driver.*

71.   Figure 12. Is Sea Level ODN the same as Total Water Level OD? Please explain why the axis labels are different. The panel indication (a), (b), and (c) are not shown.

 Yes, ODN and OD are the same datum for Great Britain. We use OD.

72.   Line 653. Please clarify which ranges are considered extreme in parenthesis.

The text has been edited to reflect the levels which are considered extreme.

73.    Lines 654–656. "The volume of riverine freshwater is the dominant driver contributing to high water levels in the estuary. This could be evidence of the backwater effect, where high river discharge can push back low levels of tidal water, resulting in a temporary increase in water levels within the estuary". Please provide some references that corroborate it.

Thank you, we would suggest the following reference:

*Feng, D., Tan, Z., Engwirda, D., Liao, C., Xu, D., Bisht, G., Zhou, T., Li, H.-Y., and Leung, L. R. (2022). Investigating coastal backwater effects and flooding in the coastal zone using a*

*global river transport model on an unstructured mesh, Hydrol. Earth Syst. Sci., 26, 5473–5491, https://doi.org/10.5194/hess-26-5473-2022.*

And

*Ikeuchi, H., Hirabayashi, Y., Yamazaki, D., Kiguchi, M., Koirala, S., Nagano, T., Kotera, A., Kanae, S. (2015). Modeling complex flow dynamics of fluvial floods exacerbated by sea level rise in the Ganges–Brahmaputra–Meghna Delta. Environ. Res. Lett. 10 124011*

74.  Lines 658–659. Please re-order the sentence "It is when the river discharge is between 450-550 m³/s in the Conwy Estuary that flood forecasts need to be particularly accurate. " to "Results shown that flood forecasts need to be particularly accurate for Conwy Estuary when the river discharge is between 450-550 m³/s". In addition, please say in parentheses if this range of values is mild or extreme.

This text has been edited in section 4.1.1.

75.   Line 679. In the sentence: "The parameter space could be developed by considering additional hydrograph time lags and exploring the timing of the surge relative to tidal high water, which could influence the magnitude and volume of the total water level (Lyddon et al., 2018; Khanam et al., 2021)." I suggest two references:

*Costa W, Bryan KR, Stephens SA and Coco G (2023) A regional analysis of tide-surge interactions during extreme water levels in complex coastal systems of Aotearoa New Zealand. Front. Mar. Sci. 10:1170756. doi: 10.3389/fmars.2023.1170756*

And

*Arns, A., Wahl, T., Wolff, C., Vafeidis, A. T., Haigh, I. D., Woodworth, P., et al. (2020). Non-linear interaction modulates global extreme sea levels, coastal flood exposure,*

*and impacts. Nat. Commun. 11, 1–9. doi: 10.1038/s41467-020-15752-5*

These references have been added to the manuscript.

76.  Line 688. "Sea-level rise and geomorphic changes will lead to a new baseline for flooding and new driver-thresholds and interactions ". Reference suggestion:

*"Khojasteh, D., Glamore, W., Heimhuber, V., and Felder, S. (2021). Sea level rise impacts Estuar. dynamics: A review. Sci. Total Environ. 780, 146470. doi: 10.1016/j.scitotenv.2021.146470"*

These references have been added to the manuscript.

77.   Lines 713–716. "The research highlighted the incomplete nature of recorded flooding extents held by national agencies, which are important to build a database of past episodes of flooding (e.g., when and where has flooded, and under what conditions) and undertake further analyses such as temporal trends in flooding. Such a database is crucial for developing accurate and timely flood warnings. ". This passage is a bit unclear; maybe change it to

"The research highlights that the recorded flooding extents held by national agencies are incomplete. This database is important to build knowledge on past flooding episodes (e.g., when and where has flooded, and under what conditions), undertake further analyses such as temporal trends in flooding, and develop accurate and timely flood warnings."

The suggested text has been used in the manuscript.

78.    Section 5. It is confusing that historic flooding records are included in the Conclusion section but not in Results. Instead, they are described in Methods. I suggest removing the historic events from the conclusion or moving the historic flooding records from methods to results.

This is addressed in response to comment n° 14.

79.  Section 5. See comment n° 14. Suggestion: the third and fourth paragraphs should be joined together and placed as the second paragraph. The paragraph in lines 713– 720 should be the third or last.

Suggested changes have been made in the manuscript.